# The fast and the frugal: Divergent locomotory strategies drive limb lengthening in theropod dinosaurs

T. Alexander Dececchi[1]*, Aleksandra M. Mloszewska[2]☯, Thomas R. Holtz, Jr.[3,4]☯, Michael B. Habib[5]☯, Hans C. E. Larsson[6]☯

1 Division of Natural Sciences, Department of Biology, Mount Marty College, Yankton, South Dakota, United States of America, 2 Independent Researcher, Sudbury, ON, Canada, 3 Department of Geology, University of Maryland, College Park, Maryland, United States of America, 4 Department of Paleobiology, National Museum of Natural History, Washington, DC, United States of America, 5 Integrative Anatomical Sciences, Keck School of Medicine of USC, University of Southern California, Los Angeles, California, United States of America, 6 Redpath Museum, McGill University, Montreal, Quebec, Canada

☯ These authors contributed equally to this work.
* alex.dececchi@mtmc.edu

**Data Availability Statement:** All relevant data are within the manuscript and its Supporting Information files.

## Abstract

Limb length, cursoriality and speed have long been areas of significant interest in theropod paleobiology, since locomotory capacity, especially running ability, is critical in the pursuit of prey and to avoid becoming prey. The impact of allometry on running ability, and the limiting effect of large body size, are aspects that are traditionally overlooked. Since several different non-avian theropod lineages have each independently evolved body sizes greater than any known terrestrial carnivorous mammal, ~1000kg or more, the effect that such large mass has on movement ability and energetics is an area with significant implications for Mesozoic paleoecology. Here, using expansive datasets that incorporate several different metrics to estimate body size, limb length and running speed, we calculate the effects of allometry on running ability. We test traditional metrics used to evaluate cursoriality in non-avian theropods such as distal limb length, relative hindlimb length, and compare the energetic cost savings of relative hindlimb elongation between members of the Tyrannosauridae and more basal megacarnivores such as Allosauroidea or Ceratosauridae. We find that once the limiting effects of body size increase is incorporated there is no significant correlation to top speed between any of the commonly used metrics, including the newly suggested distal limb index (Tibia + Metatarsus/ Femur length). The data also shows a significant split between large and small bodied theropods in terms of maximizing running potential suggesting two distinct strategies for promoting limb elongation based on the organisms' size. For small and medium sized theropods increased leg length seems to correlate with a desire to increase top speed while amongst larger taxa it corresponds more closely to energetic efficiency and reducing foraging costs. We also find, using 3D volumetric mass estimates, that the Tyrannosauridae show significant cost of transport savings compared to more basal clades, indicating reduced energy expenditures during foraging and likely reduced need for hunting forays. This suggests that amongst theropods, hindlimb evolution was not dictated by one particular strategy. Amongst smaller bodied taxa the competing pressures of being

**Funding:** The author(s) received no specific funding for this work.

**Competing interests:** The authors have declared that no competing interests exist.

both a predator and a prey item dominant while larger ones, freed from predation pressure, seek to maximize foraging ability. We also discuss the implications both for interactions amongst specific clades and Mesozoic paleobiology and paleoecological reconstructions as a whole.

## Introduction

Non-avian theropod dinosaurs were the dominant terrestrial carnivores during much of the Mesozoic. They occupied much of the available niche space [1–3], and ranged in size from <200g to approximately 9000kg [4, 5]. While no single adaptation is likely to explain such widespread dominance and diversity of form, the bipedal locomotory system employed by theropods is invoked as an important reason for the success of this lineage [6]. For animals, the speed at which they travel is a critical factor in their survival strategy as it impacts all aspects of food collection, dispersal, migration and predator avoidance [7]. Because of this, much work has been done to model locomotion and how it affects different aspects of theropod life history and behavior, such as movement efficiency, turning radius, balance [8–14]. Additionally, studies of the growth across the clade, both ontogenetically and allometrically [15–18], have shown marked difference in traditional markers for cursorial potential [14, 19], such as intralimb ratios, lower limb length and relative total limb length. This includes comparisons of derived coelurosaurian theropods at all body sizes to more basal clades, and have been suggested to be linked to a refinement of running ability in this group [8, 15, 19].

Recent work by Persons and Currie [14] attempted to further this by quantifying relative cursoriality amongst non-avian theropods through the use of the distal hind limb indices (tibia + metatarsal length/ femur). They argued that the application of this metric could identify those taxa with the highest top speed and attempted to establish that it had significant impact on the ecological role and the diversification of theropods. Yet the challenge is that much of an organisms locomotory repertoire, both in terms of percentage of behaviours and duration, is at lower speeds [20]. This is especially true in carnivores who often spend hours searching for or pursuing prey and low to moderate speeds between bursts of high speed running [21, 22]. We suggest that the importance of top speed may have been overestimated in Persons and Currie. In this study, we show that this estimate changes after considering that much of the energy budget and life history of a predator is spent at lower gears, the relative speed of predators compared to suspected prey items, locomotor energetics role in shaping the evolutionary landscape for theropods, and the effect of other factors such as body size in their analysis.

Here we re-examine locomotion in non-avian theropods, applying indices based on estimates of top speed and energetic expenditures to get a more complete sense of how differences in relative limb lengths, and components within the limb itself, reflect the paleobiology and paleoecology of these creatures. In extant vertebrates the walk to run transitions occurs at a Froude number > 1 [23], and a similar value is expected to hold for non-avian theropods with even the largest being suspected to achieve this feat [10, 11]. Following these parameters we define running ability here as the ability to achieve speeds corresponding to Froude numbers significantly higher than 1, as opposed to running capacity which is the ability to generate values of 1. We hypothesize that allometry will have significant consequences for running ability and that the selective weighting of top speed versus reducing energetic expenditures will vary across Theropoda concordant with changes in body size. We also hypothesize that amongst the largest theropods, > 1000kg, running ability, as assessed by top speed potential, will not be

a significant factor in the influencing the relative level of elongation of the distal hindlimb, including in tyrannosaurs. Our goal is to more accurately understand the selective pressures that shaped limb length and intralimb proportion evolution across theropods, and to compare how the evolution of extremely body size, greater than 1000 kg, may have altered the drivers for distal limb indices. Through this we seek to more accurately reconstruct patterns of cursoriality and foraging strategies amongst theropods and understand better how they shaped the ecosystems in which they lived.

## Materials

### Relative leg length and max speed

Measurements of snout to vent length along with hindlimb lengths for 93 specimens for 71 different genera of avian and non-avian theropods were collected from the literature and personal measurements (S1 Table) that had at least one hindlimb element recorded. This included 82 specimens with a complete hindlimb preserved such that leg length and hip height could be estimated and from that speed calculated (S2 Table). Sampling includes members form all major clades and multiple specimens per species, often from different ontogenetic stages, where possible to capture the maximum diversity of Mesozoic theropod hindlimb disparity. These taxa range in SVL size from 70 to over 5000 mm. A subset of this data, 22 non-avian theropods where femoral circumferences were available, were selected to examine how body mass relates to various metrics of leg elongation (S3 Table). We only included non-avian theropods as previous work has suggested significant allometric and functional shifts in the limbs of early avians compared to their non-avian ancestors [15]. This dataset was then expanded to 77 specimens by including multiple taxa without SVL measurements to capture more non-avian theropod diversity (S4 Table). We chose several different metrics to evaluate the connection between hindlimb length and speed including total hindlimb length, distal hindlimb index and hindlimb/ SVL, hindlimb length / $m^{1/3}$ and metatarsal length $m^{1/3}$.

To calculate maximal running speed, we used several estimators to be able to compare values across proxies. The first is Froude number, which is a dimensionless number that allows for relative size to be removed from velocity calculations [24]. The second is locomotor velocity, which is calculated as [10]

$$V = \sqrt[2]{Fr} \div h \times g.$$

Where v is the velocity in m/s, Fr = Froude number, h = hip height taken here as hindlimb length and g is gravity. For hip height we chose to estimate it as 0.8 x total limb length, which corresponds to the level of crouch seen in large terrestrial modern birds [25] and mimics the values seen in other similar studies [10]. For taxa with a body mass of less than 1000kg we calculated a range of Froude values from 0.25 to 15 to document behaviors from slow walk to top speed while being within the range possible for both non-avian and avian theropods [11, 26–30]. For larger taxa a maximum Froude number of 5 was used as this is suspected to be the limit that they could achieve [10]. In addition to using Froude number we also calculated maximum speed based on another methodology: the stride length based on either Alexander [31]

$$V = 0.25g^{0.5}\lambda^{1.67}h^{-1.17}$$

Or the correction by Ruiz [32]

$$V = 0.226g^{0.5}\lambda^{1.67}h^{-1.17}$$

Where λ is the relative stride length (RSL). We took for RSL values of 2 and 4.5 corresponding to a slow and a fast burst run and within the range seen in theropod trackways [27, 33]. Finally, we used either published 3-D volumetric estimates of body mass,[9, 34, 35] or generated estimates based on femoral circumference [36] to calculate top speed based on mass limiting factor [7]. This last value is grounded on extant taxa and suggests mass specific limitations on acceleration and top speed, something seen in the modern realm but not factored in by our other metrics.

**Postural considerations.** Our selection of a constant postural position raises the possibility that allometric changes in crouch level could have effects on some of our calculations. Allometric changes in posture are known to occur in mammals, with the most upright stance occurring between 100-300kg depending on the clade [37–39] and birds [25, 40], especially during running. Small bodied modern avians display a highly "crouched" body plan and thus a "groucho walk" style of locomotion that is fundamentally different than seen in non-avian theropods [41]. While the exact location of this postural shift is unknown, it is suspected to be not a distinct shift from "hip based" to "knee based" but a more gradual transition [42]. Recent work suggests that, based on the center of mass, this shift may have started as early as with basal theropods [34], though this study does not see a significant dorsoventral shift occurring until Neornithes (crown-group birds). As such, we also ran a permutation using a variable hip height based on the differences between leg length (including the phalanges) and hip height seen across a sample of extant ground birds while running at moderate speeds [40]. This data overlaps with that of [25], producing similar values. Based on this dataset, all extant birds greater than 5kg in mass relative hip height is at least 74% of total leg length [40] and regression analysis suggests that, for fossil theropod taxa greater than 40 kg in mass, values are minimally 80%. If we used only the length of the femur, tibia and metatarsus this this value is 80% or above for all species except the smallest (painted quail) whose mass is smaller than non-avian theropod investigated here at only 31g. This is similar to more recent findings by Bishop et al. [43], which found degree of crouching (the difference between hip height and leg length as defined by them to only include the femur, tibia and metatarsus) at less than 20% in species with masses above 0.6kg. Given that the level of crouch is suspected to be higher in neornithines than in non-avian theropods due to changes in running style, these should be considered lower bound estimates in terms of effective hip height, meaning our 0.8HL estimate is similar to what would be expected even in small bodied theropods and after accounting for allometric shifts in posture. Furthermore, as the allometric effects when including foot length into leg length, converge with our previous estimate at ~40kg using [40] and less than 1kg using [43] as well as the fact that mammalian work across groups indicates that upright postures are present in all taxa greater than 300kg [38], any postural effects would not be expected to alter our energetic analysis of large bodied (>600kg) theropods.

## Energy consumption in large theropods

Estimates of energetic expenditure during foraging amongst the largest bodied theropods were calculated from mass estimates based on several published 3-D body volume reconstructions, one of the most a reliable and comparable method to estimate body mass [9, 10, 34, 35] (Table 1, S6 Table). The advantages of using 3D volumetric mass estimates to compare between taxa is that it is an estimate generated using the an internally consistent, replicated and validated methodology tailored for individual specimens and does not rely on hindlimb dimension, which we are using in other aspects, thus preventing circularity in our arguments. This allows us to compare relative elongation levels with an outside proxy and one that reduces the potential issue that even in the most constrained size reconstruction from femoral length

**Table 1. Mass, hindlimb and hip height and cost of transport for 3D volumetric specimens examined here.**

| Source | Taxon | Specimen | Mass (kg) | HL (mm) | HH (cm) | CoT |
|---|---|---|---|---|---|---|
| Bates et al. 2009 | *Struthiomimus* | BHI 1266 | 423 | 1800 | 144.0 | 1.97 |
| Bates et al. 2009 | *Allosaurus* | MOR 693. | 1500 | 1699 | 135.9 | 2.06 |
| Bates et al. 2009 | *Tyrannosaurus* | MOR 555/USNM 555000 | 6072 | 3036 | 242.9 | 1.31 |
| Bates et al. 2009 | *Acrocanthosaurus* | NCSM 14345 | 6177 | 2675 | 214.0 | 1.45 |
| Bates et al. 2012 | *Tyrannosaurus* | BHI 3303 | 7655 | 3196 | 255.7 | 1.26 |
| Pontzer et al. 2009 | *Archaeopteryx* | MB.Av.101 | 0.25 | 158 | 12.6 | 12.80 |
| Pontzer et al. 2009 | *Marasuchus* | Composite | 1.00 | 170 | 13.6 | 12.10 |
| Pontzer et al. 2009 | *Microraptor* | IVPP V13352 | 1.20 | 291 | 23.3 | 8.00 |
| Pontzer et al. 2009 | *Compsognathus* | BSP AS I563 | 3.00 | 209 | 16.7 | 10.32 |
| Snively et al. 2018 | *"Raptorex"* | LH PV18 | 47 | 998 | 79.8 | 3.10 |
| Snively et al. 2018 | *Eustreptospondylus* | OUM J13558 | 206 | 1245 | 99.6 | 2.61 |
| Snively et al. 2018 | *Dilophosaurus* | UCMP 37302 | 372 | 1412 | 113.0 | 2.37 |
| Snively et al. 2018 | *Gorgosaurus* | TMP 91.36.500 | 496 | 1825 | 146.0 | 1.95 |
| Snively et al. 2018 | *Tyrannosaurus* | BMRP 2002.4.1 | 660 | 2120 | 169.6 | 1.73 |
| Snively et al. 2018 | *Ceratosaurus* | USNM 4735 | 678 | 1429 | 114.3 | 2.35 |
| Snively et al. 2018 | *Gorgosaurus* | AMNH 5664 | 688 | 1928 | 154.2 | 1.87 |
| Snively et al. 2018 | *Tarbosaurus* | ZPAL MgD-I/3 | 727 | 1845 | 147.6 | 1.93 |
| Snively et al. 2018 | *Allosaurus* | USNM 4734, UUVP 6000 | 1512 | 1985 | 158.8 | 1.82 |
| Snively et al. 2018 | *Allosaurus* | MOR 693 | 1683 | 1795 | 143.6 | 1.97 |
| Snively et al. 2018 | *Yangchuanosaurus* | CV 00215 | 2176 | 1988 | 159.0 | 1.82 |
| Snively et al. 2018 | *Sinraptor* | ZDM 0024 | 2374 | 2340 | 187.2 | 1.61 |
| Snively et al. 2018 | *Gorgosaurus* | AMNH 5458 | 2427 | 2640 | 211.2 | 1.46 |
| Snively et al. 2018 | *Gorgosaurus* | NMC 2120 | 2427 | 2634 | 210.7 | 1.47 |
| Snively et al. 2018 | *Tarbosaurus* | PIN 552–1 | 2816 | 2415 | 193.2 | 1.57 |
| Snively et al. 2018 | *Acrocanthosaurus* | NCSM 14345 | 5474 | 2676 | 214.1 | 1.45 |
| Snively et al. 2018 | *Giganotosaurus* | MUCPv-CH-1 | 6908 | 3020 | 241.6 | 1.32 |
| Snively et al. 2018 | *Tyrannosaurus* | CM 9380 | 6987 | 3124 | 249.9 | 1.29 |
| Snively et al. 2018 | *Tyrannosaurus* | FMNH PR 2081 | 9131 | 3261 | 260.9 | 1.24 |
| Persons and Currie 2014 | *Khaan* | MPC-D 100/1127 | 5 | 391 | 31.3 | 6.37 |
| Persons and Currie 2014 | *Velociraptor* | MPC1—/986 | 15 | 592 | 47.4 | 4.63 |
| Persons and Currie 2014 | *Ajancingenia* | MPC-D 100/30 | 17 | 634 | 50.7 | 4.39 |
| Persons and Currie 2014 | *Ornithomimus* | TMP 95.11.001 | 150 | 1220 | 97.6 | 2.65 |
| Persons and Currie 2014 | *Gorgosaurus* | TMP 91.36.500 | 400 | 1815 | 145.2 | 1.95 |
| Persons and Currie 2014 | *Tyrannosaurus* | BHI 3303 | 5622 | 3196 | 255.7 | 1.26 |

Specimens used for costs of transport analysis based on published 3D volumetric data. HL stands for hindlimb length which here is taken as femur + tibia + central metatarsal lengths. HH is hip height, calculated as 0.8*HL. CoT is cost of transport, see text for details.

or circumference has the confidence intervals that can span orders of magnitude in mass, thus making it extremely difficult to determine if two taxa, even if they show similar values for said estimator, are actually similar in body mass [36]. Since this is critical for this type of analysis, using more generalized body mass figures obtained through femoral circumference as used in our previous section (S3–S5 Tables) would be inadequate for the purposes used here. We attempted, whenever possible, to confine our analysis to looking at taxa only with volumetric mass estimates taken from within the same study to ensure that differences in volumetric estimations methods do not cause spurious results. When multiple masses were presented we chose to run our analysis using the "best estimate" model as defined in the original papers for

all specimens as opposed to the heaviest or lightest. This was done so that we could get direct comparative data using the same variables and not bias the results by artificially inflating or reducing the body mass volumes a priori. This model often produces very similar mass estimates for these two specimens, such as between *Tyrannosaurus* and *Acrocanthosaurus* using which differs only by 1.7% [35].

As a significant proportion of a predator's daily activity budget is occupied by foraging [21, 22, 44, 45] we chose to reconstruct the energetics values based on cost of foraging [10] using an absolute speed of 2 m/s to simulate a slow walk (Froude number <0.25). This is similar to the estimate used by [45] and well within the walking range estimated from known trackways for large theropods [32, 46–51]. We then combined cost of transport (CoT) with a speed of transport to calculated the costs to forage to both cover a set distance (18 km, the daily expected foraging range of a large theropod [44] and 6570km which is the yearly total) as well as over a series of time intervals (12 hours/ day foraging per [45] and 1 year) to examine the difference in expenditures across comparable sized taxa.

Finally, to compare proportional expenditures we performed two different analyses. First we transformed the difference calculated in kilojoules (kj) into kilograms (kg) of meat by using the energetic conversion values from [52] for large mammalian carnivores. While we understand that the digestive and excretory methods of theropods make it difficult to estimate of the amount of meat required, especially if they excreted uric acid like modern avians which leads to greater energy loss [52], regardless these parameters are similar to previous studies [45] and defendable based on suspected aerobic capability [10]. In addition, previous work has suggested that the largest theropods would have a metabolic rate equivalent of a 1000 kg carnivorous mammal [53], which is approaching the theoretical maximum size for a terrestrial carnivore [21]. We also compared expenditure values to estimates of Basal metabolic rate based on the equations of McNab [54] and Grady et al. [55]. This allows us to remove the effect of potential digestive absorptive differences between macrocarnivourous mammals and theropods from our data. Regardless of whether these taxa were true truly endotherms or mesotherms, these values should produce reasonable estimates of relative disparities in expenditures to compare between specimens.

## Results

### Relative leg length

We observe a correlation between the relative hindlimb versus distal hindlimb indices, except in in small to medium sized theropods less than 1200 mm SVL, where we observe a disconnect between these variables (S1 Table). This is especially clear when comparing some contemporaneous taxa, such as compsognathids and microraptorines (Fig 1). The former has been suggested to be highly cursorial while the latter were not based on evaluation of the hindlimb index alone [14]. Our results dispute this finding, as well as previous work that ignored allometric effects in comparing smaller compsognathids to mid-sized and larger dromaeosaurs such as *Velociraptor* or *Deinonychus* [14]. Focusing on non-avian theropods, as avian theropods have different hindlimb scaling factors compared to non-avian theropods [15], we also see negative allometric scaling in intralimb ratios which alone could influence comparison between these two clades. Even at similar sizes the divergence between relative hindlimb versus distal limb metrics is clearly illustrated by comparing the Yixian biota contemporaries *Changyuraptor* and *Sinosauropteyrx*, both of which are suspected small carnivores that differ in length by 10mm (~ 2% total SVL). *Changyuraptor* show a relative hindlimb index of 0.96, which is significantly higher than that is seen in *Sinosauropteyrx* (0.57) while showing a distal limb index value 8% lower. This pattern of high hindlimb indices whiles showing relatively

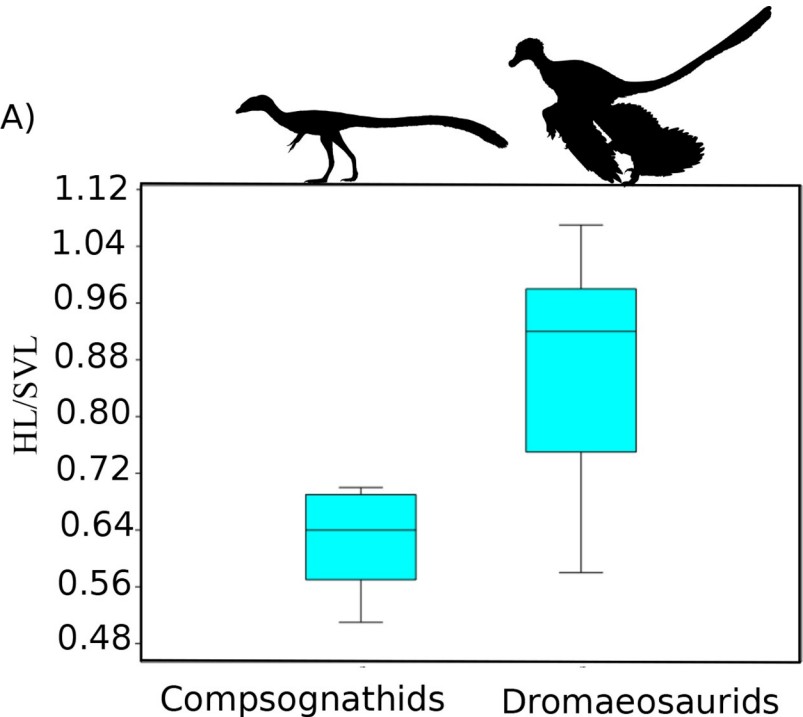

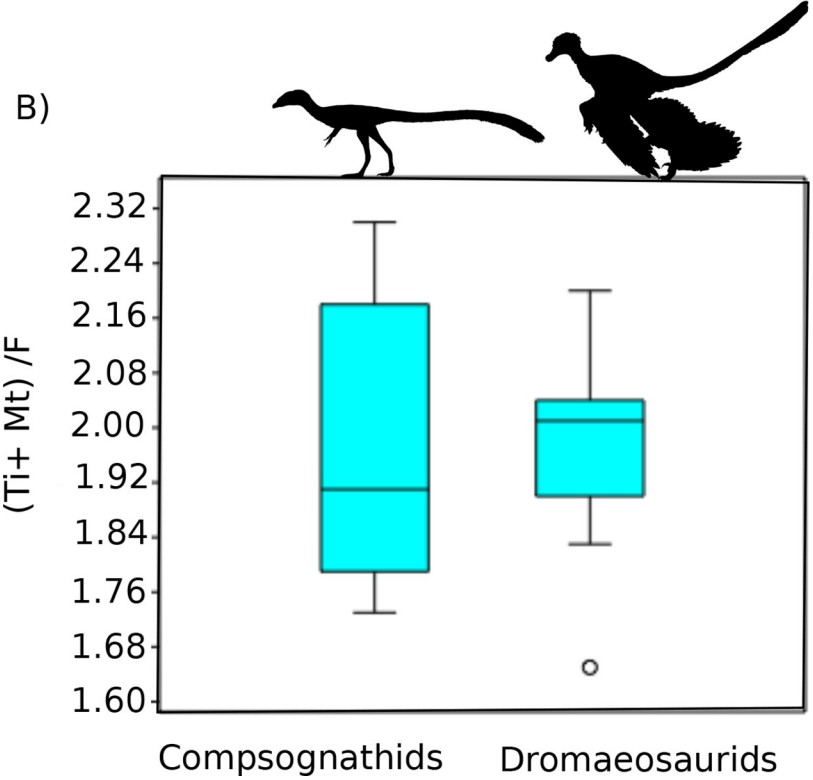

**Fig 1. Hindlimb index comparisons between compsognathids and dromaeosaurids.** Comparison of two small bodied theropod clades, the compsognathids and dromaeosaurids, using different hindlimb indices purported to be associated with cursorial ability. Note the significant difference in how the running ability and top speed would be reconstructed depending on the metric selected. Using relative hindlimb length (A) we find a significant difference between the two groups (unequal variance t-test 5.1471, p = 0.001) and would reconstruct dromaeosaurids as significantly faster than compsognathids. Using distal limb index (B) we see no difference between clades (unequal variance t-test 0.7713, p = 0.45). Silhouette modified from those in Phylopic image repository (Phylopic.org) created by Joh Conway and Brad McFeeters.

mild distal limb values is seen across all small bodied microraptorines and basal troodontids, with the opposite trend seen in small bodied compsognathids and basal birds. Interestingly, amongst anchiornithids (a clade of small bodied paravians who have recently been suggested to be more closely related to birds than either dromaeosaurids or troodontids [56, 57]) we see a diverse pattern of values ranging from *Anchiornis* at the high end (92–95%) to *Caihong* (78%) at the lower end, though the latter is still similar to what is seen in the cursorial oviraptorosaur *Caudipteryx* (0.75–0.79). We calculated maximum speed potential, using our derived hip heights and calculating stride length based on 4 times hip height as well as at Fr of 15, with the larger hindlimb indices in many microraptorines allowing them to achieve higher top end speed suggesting a sharp demarcation between burst speed potential between microraptorines, contemporaneous small bodied compsognathids and basal birds (Fig 2, S2 Table). Depending on the speed estimator used *Changyuraptor* shows top speed between 5.13–7.98 m/s which is 1.2–1.9 m/s (4.9–6.9 km/hr) higher than *Sinosauropteryx* using the same metric. Adjustments in hip height produced lower absolute speeds but the relative differences between these taxa remains unaffected. While there is much variation between speed estimators, for each one we see *Changyuraptor* display a top speed that is 130+% higher than *Sinosauropteyrx*. Similar results are seen between other parings (Fig 2, S2 Table). We also find that the juvenile tyrannosaur "*Raptorex*" (a suspected young *Tarbosaurus* specimen [58]), shows significant burst speed potential, even higher than similar sized ornithomimids, oviraptorosaurs or basal tyrannosauroids (S2 Table). This supports the idea that juvenile tyrannosaurids were highly cursorial [9, 13, 14].

To evaluate the general applicability of these various hindlimb ratios across Theropoda as a good proxy for top speed, we compared distal limb index, hindlimb index as well as metatarsal and whole leg length compared to body size using top speeds at FR = 5 and under the mass limiting top speed equation of [7]. Using the primary dataset (S3 Table) we find that all proxies have relatively low correlation value, with distal limb index ($r^2 = 0.55$) as the only metric showing a significant correlation when using speed based on Froude number. When we take into account the limiting factor of increasing body mass, all metrics show precipitous decrease in correlation value with none of them showing a significant relationship to speed (Fig 3, S3 Table). To confirm that this was not due to the taxon sampling we used our larger dataset (S4 Table), which though it did not allow us to evaluate HL/SVL, did allow for testing the other three metrics. Using just Froude number all three metrics showed significant correlations to top speed, with distal limb index showing the highest correlation ($r^2 = 0.48$) (Fig 4, S5 Table). Once again, when correcting for mass all three metric correlations drop to insignificant levels, with distal limb index showing a correlation coefficient of less than 0.04.

**Energy consumption in theropods.** To determine whether the greatest selection pressure for hindlimb elongation was a savings in terms of transport costs or maximizing top speed, we compared top speeds calculated using Fr = 5 or 15 to that accounting for body mass in our expanded limb dataset (Fig 4, S4 Table). Across all speed estimates we find that at lower size classes the estimated top speed is lower than the theoretical maximum generated through [7]. However, this changes in mid to large size theropods. Depending on the speed estimator used

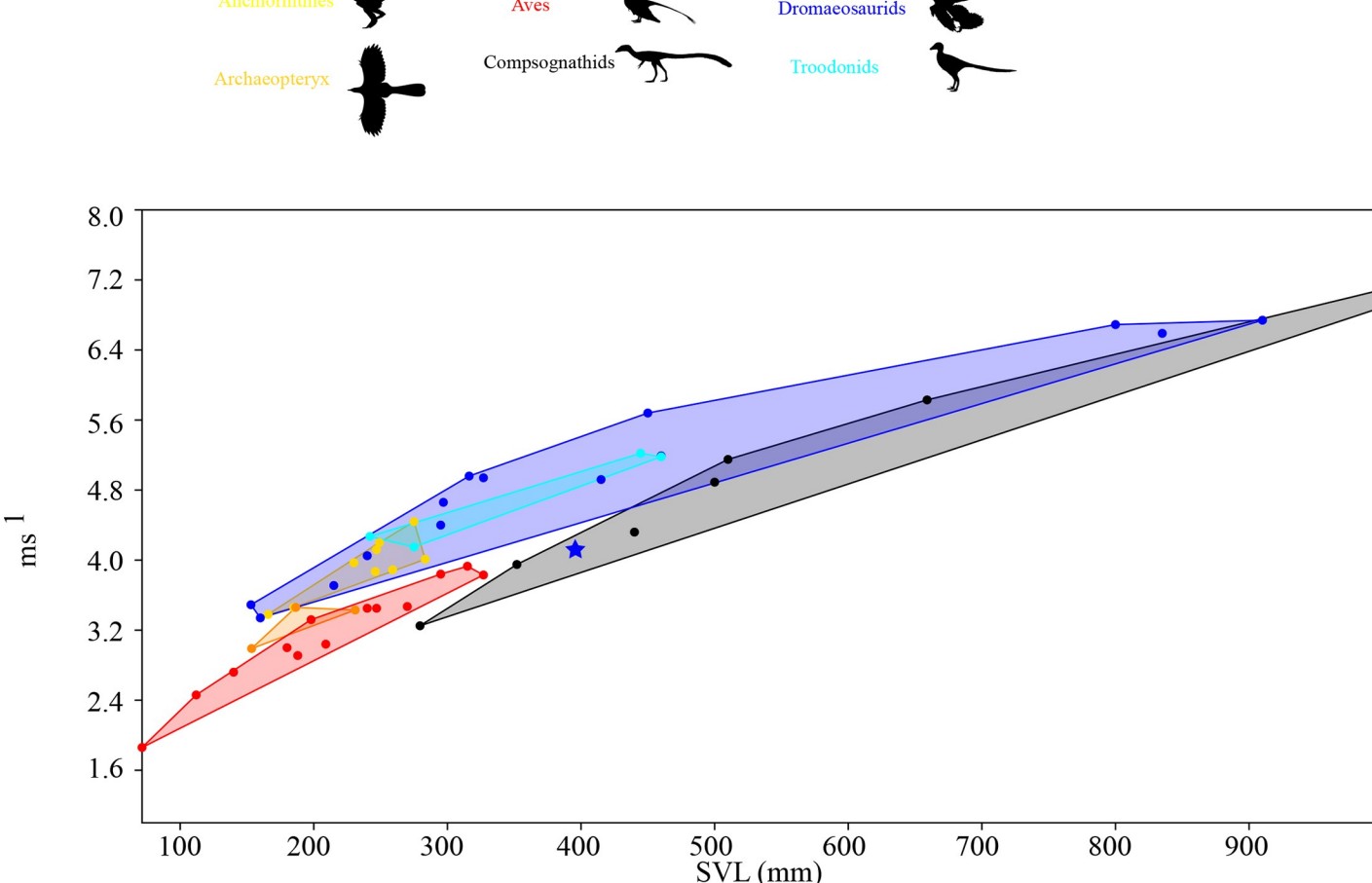

**Fig 2. Comparisons of maximum speed potential between small coelurosaur clades.** Top speed comparison between clades using speed calculated from equations in [28], though all reconstruction methods show similar patterns. The blue start represents *Halszkaraptor*, whose position is likely based on its proposed unique semi aquatic lifestyle. Of note, at lower speeds the dromaeosaurs, more specifically microraptorines, show distinctly higher top speed than comparably size compsognathids, *Archaeopteryx* specimens or basal birds and similar values to troodontids. Silhouette modified from those in Phylopic image repository (Phylopic.org) created by Joh Conway, Matt Martynuik, Gareth Monger and Brad McFeeters.

the body mass limiting top speed drops below the others at around 500 kg using a Fr of 15 and 2000kg using a Fr = 5. This corresponds to a hip height of ~ 1.5–2.1 m.

Using existing volumetric derived masses, we calculated the cost of transport [10] across a range of theropods and dinosauriforms from 0.25 kg to greater than 9000kg (Table 1). Our results show that, among the large bodied theropods, tyrannosauroids show a significantly lower cost of transport than comparable size more basal taxa, with differences most exacerbated in juvenile and sub-adult size classes (Tables 1 and 2). If we hold velocity constant at 2 m/s, we see significant differences in energetic values between tyrannosauroids and other large taxa (Table 2), based on the relative elongation of their hindlimbs. In order to assess what level of difference in foraging efficacy in terms of CoT make in terms of overall energy expenditure we reconstructed daily energy expenditure budgets for tyrannosaurids and more basal theropods that differed from each other by less than 3% of total body mass. While the differences in the cost of transport values between tyrannosauroids and other large theropods may appear minimal, ranging from only 0.03–0.62 j/kgm, when they are evaluated for taxa at these large sizes and over longer temporal durations they produce significant differences (Table 3A and

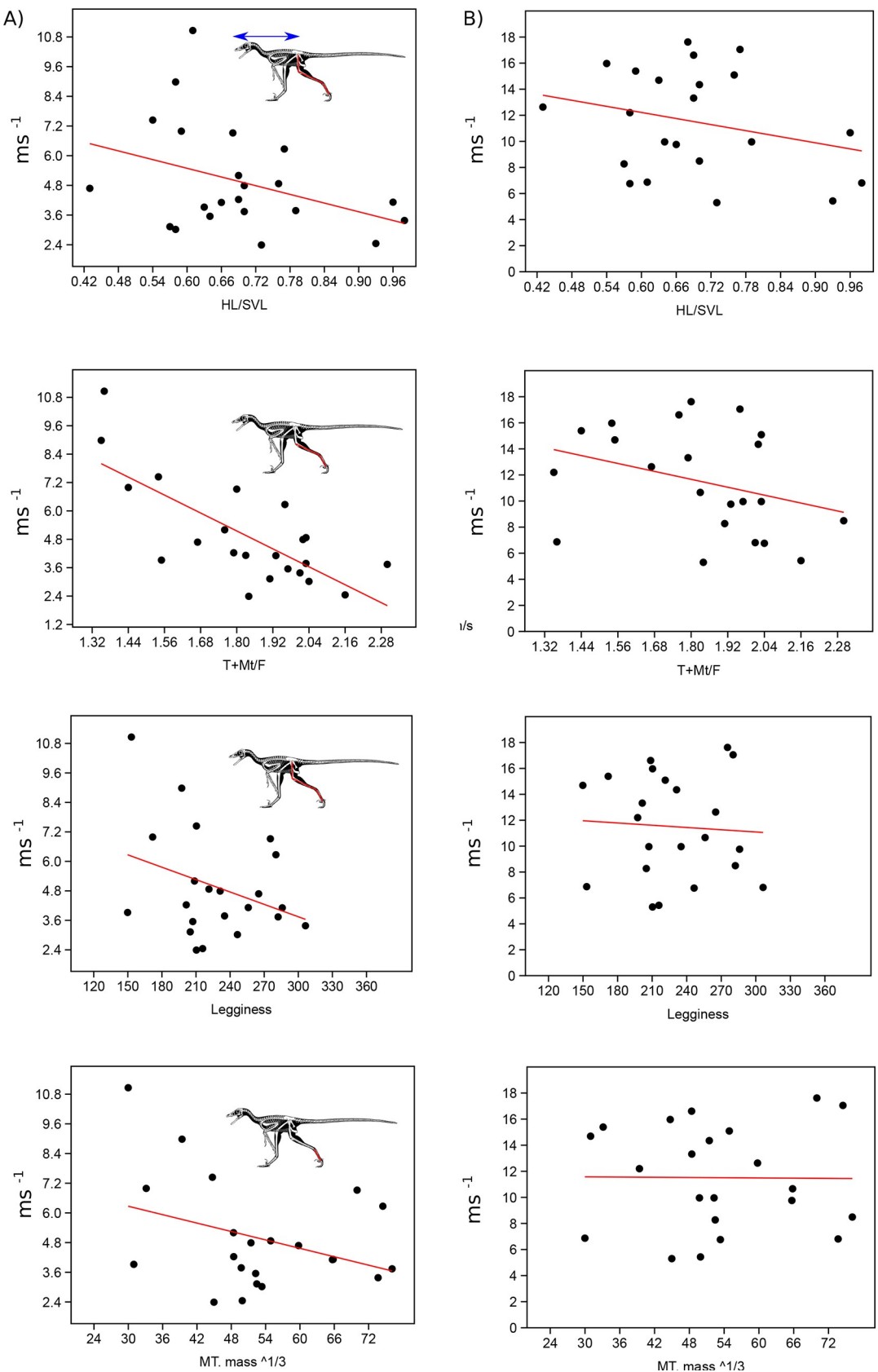

**Fig 3. Comparing proxies to running speed estimates using SVL only database.** Evaluation of the fit of hindlimb index proxies to estimated top speed at Froude = 5 (A) and (B) using the mass induced limitation as proposed in [7] using the primary dataset of taxa with SVL data. HL/SVL = total hindlimb length/ snout to vent length, T+Mt/F = tibia+ longest metatarsal length / femoral length, Legginess = hindlimb length/ body mass (kg)^1/3, MTmass^1/3 = metatarsal length/ mass body mass (kg) ^1/3. Skeletal image of *Microraptor* modified from the illustrations of S. Hartman.

3B, S1 and S2 Figs). In looking simply at how CoT values differ between comparable medium and large theropods (Table 2) vs the differences in maximum running speed implied by [7] we find the former range from 2.4–8+% while the later are less than 0.5% (S6 Table, S1 and S2 Figs). This suggests that the variation due to differences in absolute leg length are more impactful in CoT calculations than speed estimates. To further explore this we chose to look at both basal metabolic rate (BMR) and BMR + foraging costs to gain a baseline to compare relative differences in energy use. This was done to ensure we would not produce an exaggeration of the differences between taxa as, for example, the estimated daily caloric intake according to BMR using [55] for the 660 kg juvenile tyrannosaurid "Jane" is only 2400 calories or about the same as the lead author.

We see significant differences between tyrannosaurids and more basal large theropods, using either BMR or BMR + energetic expenditures for both the hourly and distance based foraging ranges. Using a 12 daily hour foraging regime per [45] we find foraging savings between similar sized tyrannosaurids and more basal forms is between 10% of daily to 300% of daily BMR (Tables 2 and 3A). We contend this suggests that this metric may be too low a baseline. Using BMR + energetic expenditure values we find differences drop, but the trends remain similar. Differences in total daily expenditure range from 1.3% in the largest *Tyrannosaurus* specimens compared to *Giganotosaurus* up to 35% when comparing the juvenile *Tyrannosaurus* "Jane" to a *Ceratosaurus* (Table 3B). This translates to between 2-16kg of extra meat a day. Interestingly, the highest values are seen when comparing *Acrocanthosaurus* (NCSM 14345) to the "Wankel" *Tyrannosaurs* specimen (MOR 555 [currently USNM 555000 with the transfer of the specimen to the National Museum of Natural History]), but is lower in the largest specimens examined here.

Given the uncertainty on the percentage of the day spent foraging, using distance traveled may provide us with a more robust comparison. Adult tyrannosaurids have been estimated to travel perhaps 18 km per day in foraging [44] which at 2 m/s would correspond to 2.5 hrs of foraging time, comparable to that seen in modern large terrestrial mammalian carnivores [21]. Over the course of a year this would amount to large bodied theropods traversing over 6500 km. If we examine distance traveled we see lower, but still significant, differences in energy expenditure ranging from 0.9 to 19.8% of total expenditure over that distance (Table 3). While these differences, around 1% in the largest theropods, may seem insignificant of the course of a year they are the equivalent of 3–6 days of total energetic expenditures (BMR + daily foraging of 18km). If we translate that to how many meals over the course of a year's foraging, it translates to over 170kg of less meat consumed in the largest specimens. This corresponds to the size of a *Ornithomimus* or subadult *Thescelosaurus* [44] and up to 1250 kg in the "Wankel" specimen compared to *Acrocanthosaurus* which is the equivalent of 5 *Thescelosaurus*.

## Discussion

### Getting up to speed

We find that using single, simple limb metrics, especially distal limb ratios, directly in judging the "cursoriality" of taxa across Theropoda is not defensible unless supplemented with other means of support. If looking at comparable sized individuals, particularly amongst small

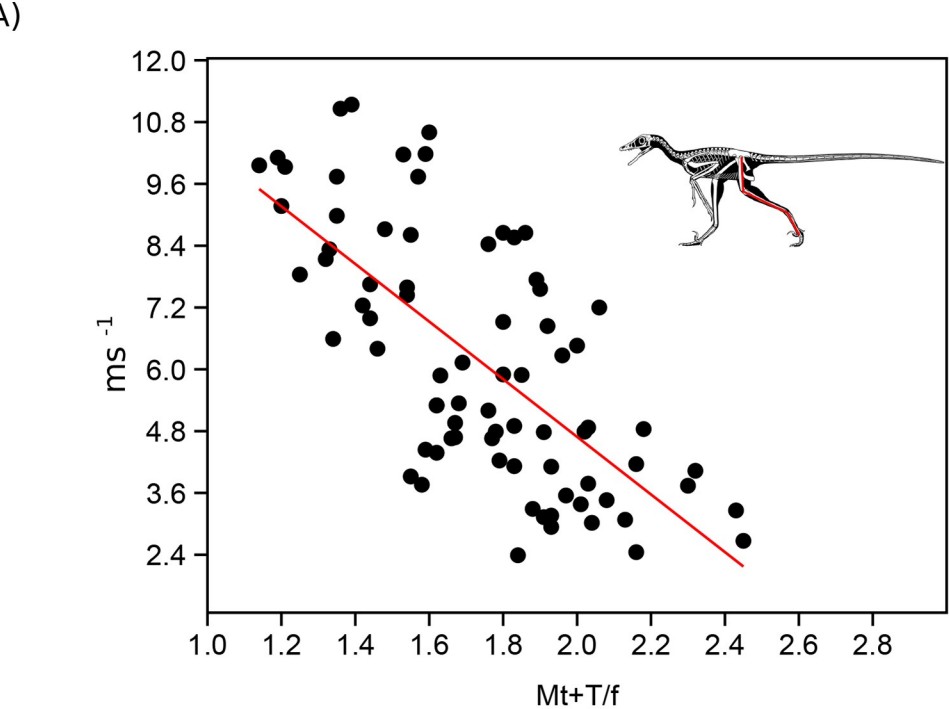

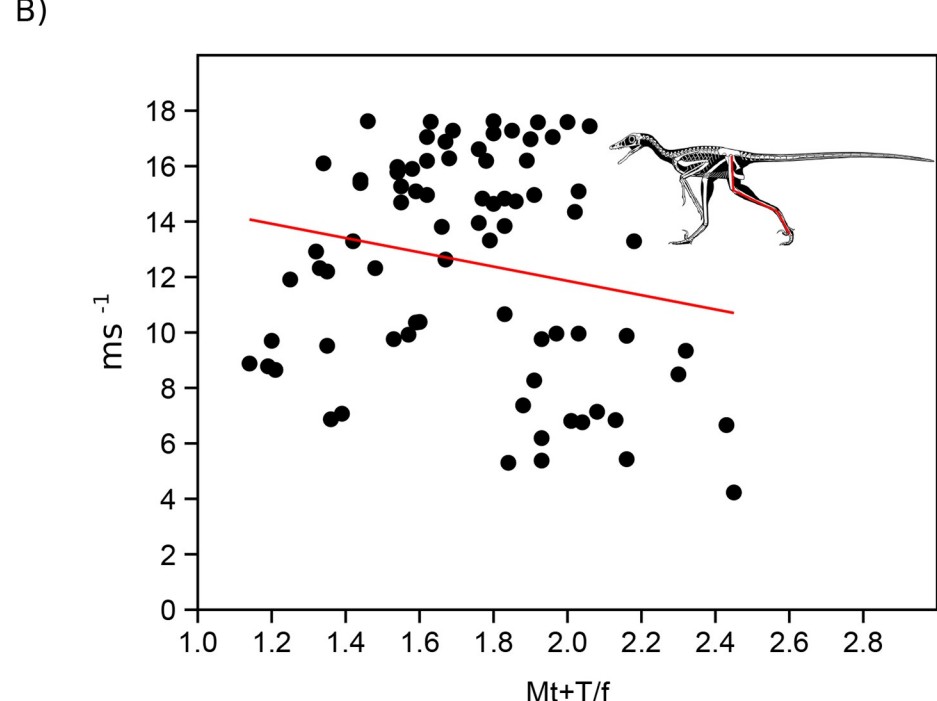

**Fig 4. Comparing proxies to running speed estimates using larger hindlimb database.** Evaluation of the fit of the distal limb index proxy to estimated top speed at Froude = 5 (A) and using the mass induced limitation as proposed in [7] in the larger hindlimb dataset (S4 Table). Skeletal image of *Microraptor* modified from the illustrations of S. Hartman.

**Table 2. Foraging costs for theropod dinosaurs.**

A)

| Taxon | specimen | Mass (kg) | HH (cm) | CoT | Foraging 12 hrs/ day | | |
|---|---|---|---|---|---|---|---|
| | | | | | hr | day | year |
| *Tyrannosaurus* | MOR 555 | 6072 | 242.9 | 1.31 | 5.75E+04 | 6.90E+05 | 2.52*10^8 |
| *Acrocanthosaurus* | NCSM 14345 | 6177 | 214.0 | 1.45 | 6.45E+04 | 7.74E+05 | 2.82E+08 |
| *Tyrannosaurus* | BHI 3303 | 7655 | 255.7 | 1.26 | 6.97E+04 | 8.36E+05 | 3.05E+08 |
| *Tyrannosaurus* | BMRP 2002.4.1 | 660.23 | 169.6 | 1.73 | 8.24E+03 | 9.89E+04 | 3.61E+07 |
| *Sinraptor* | ZDM 0024 | 2373.5 | 187.2 | 1.61 | 2.75E+04 | 3.29E+05 | 1.20E+08 |
| *Gorgosaurus* | AMNH 5458 | 2427.3 | 211.2 | 1.46 | 2.56E+04 | 3.07E+05 | 1.12E+08 |
| *Gorgosaurus* | NMC 2120 | 2427.3 | 210.7 | 1.47 | 2.56E+04 | 3.08E+05 | 1.12E+08 |
| *Tarbosaurus* | PIN 552–1 | 2816.3 | 193.2 | 1.57 | 3.18E+04 | 3.82E+05 | 1.39E+08 |
| *Acrocanthosaurus* | NCSM 14345 | 5474.1 | 214.1 | 1.45 | 5.71E+04 | 6.85E+05 | 2.50E+08 |
| *Giganotosaurus* | MUCPv-CH-1 | 6907.6 | 241.6 | 1.32 | 6.57E+04 | 7.88E+05 | 2.88E+08 |
| *Tyrannosaurus* | CM 9380 | 6986.6 | 249.9 | 1.29 | 6.47E+04 | 7.76E+05 | 2.83E+08 |
| *Tyrannosaurus* | FMNH PR 2081 | 9130.87 | 260.9 | 1.24 | 8.18E+04 | 9.82E+05 | 3.58E+08 |
| *Tyrannosaurus* | BHI 3303 | 5622 | 255.7 | 1.26 | 5.12E+04 | 6.14E+05 | 2.24E+08 |
| B) | | | | | | | |
| Taxon | specimen | mass (kg) | per km | 18 km | 6570 km | | |
| *Tyrannosaurus* | MOR 555 | 6072 | 7983 | 1.44E+05 | 5.25E+07 | | |
| *Acrocanthosaurus* | NCSM 14345 | 6177 | 8953 | 1.61E+05 | 5.88E+07 | | |
| *Tyrannosaurus* | BHI 3303 | 7655 | 9675 | 1.74E+05 | 6.36E+07 | | |
| *Tyrannosaurus* | BMRP 2002.4.1 | 660.23 | 1145 | 2.06E+04 | 7.52E+06 | | |
| *Sinraptor* | ZDM 0024 | 2373.5 | 3814 | 6.86E+04 | 2.51E+07 | | |
| *Gorgosaurus* | AMNH 5458 | 2427.3 | 3554 | 6.40E+04 | 2.33E+07 | | |
| *Gorgosaurus* | NMC 2120 | 2427.3 | 3560 | 6.41E+04 | 2.34E+07 | | |
| *Tarbosaurus* | PIN 552–1 | 2816.3 | 4416 | 7.95E+04 | 2.90E+07 | | |
| *Acrocanthosaurus* | NCSM 14345 | 5474.1 | 7932 | 1.43E+05 | 5.21E+07 | | |
| *Giganotosaurus* | MUCPv-CH-1 | 6907.6 | 9119 | 1.64E+05 | 5.99E+07 | | |
| *Tyrannosaurus* | CM 9380 | 6986.6 | 8986 | 1.62E+05 | 5.90E+07 | | |
| *Tyrannosaurus* | FMNH PR 2081 | 9130.87 | 11362 | 2.05E+05 | 7.46E+07 | | |
| *Tyrannosaurus* | BHI 3303 | 5622 | 7105 | 1.28E+05 | 4.67E+07 | | |

Foraging coasts amongst large bodied theropods based on volumetric reconstructions. A) costs in Kj on an hourly, daily and yearly basis. B) Costs of foraging in Kj per unit distance assuming a 18km daily foraging distance as per Carbonne et al (2011) for 1Km, 1 day (18 Km) and 1 year (6570Km).

theropods less than 500kg, using either HL/SVL or distal limb indices has the potential to allow for accurate assessment of relative level of cursoriality between specimens, but given the low correlation value generated in our analysis, caution is advised on using these as central pillars in paleoecological reconstructions. One major reason for this is that some indices, such as HL/SVL, are highly influenced by allometry. HL/SVL amongst non-avian theropods shows a strongly negative scaling with body size (log HL = 0.85293+/- 0.022505* log SVL+0.26446 +/- 0.063007, $r^2$ = 0.96, p(uncorr)>0.001, n = 77). Thus larger animals, up until they hit the boundary where body size limits speed and acceleration potential [7], will have higher absolute speeds due to their absolutely longer leg length. Thus, at the same Froude number, they will have higher top speed regardless of the proportions of the limb. For example, *Eustreptospondylus* (Hl = 1209 mm, HL/SVL 0.58, Distal limb index 1.43) has a higher top speed at a Froude of 5 (7.7 m/s) than *Changyuraptor* (4.6 m/s, Hl = 433 mm, HL/ SVL = 0.96), Distal limb index 1.83. Distal limb index also shows this pattern of negative allometry, though the correlation is

**Table 3. Total daily energy expenditure estimates for medium and large theropods.**

A)

| Taxon | specimen | mass (kg) | CoT | Foraging | BMR [45] | BMR [46] |
|---|---|---|---|---|---|---|
| *Tyrannosaurus* | BMRP 2002.4.1 | 660 | 1.73 | 9.89E+04 | 2.03E+04 | 1.02E+04 |
| *Ceratosaurus* | USNM 4735 | 678 | 2.35 | 1.38E+05 | 2.07E+04 | 1.05E+04 |
| *Gorgosaurus* | AMNH 5664 | 688 | 1.87 | 1.11E+05 | 2.09E+04 | 1.06E+04 |
| *Sinraptor* | ZDM 0024 | 2374 | 1.61 | 3.29E+05 | 5.19E+04 | 2.92E+04 |
| *Gorgosaurus* | AMNH 5458 | 2427 | 1.46 | 3.07E+05 | 5.28E+04 | 2.97E+04 |
| *Gorgosaurus* | NMC 2120 | 2427 | 1.47 | 3.07E+05 | 5.28E+04 | 2.97E+04 |
| *Acrocanthosaurus* | NCSM 14345 | 5474 | 1.45 | 6.85E+05 | 9.60E+04 | 5.79E+04 |
| *Tyrannosaurus* | BHI 3303 | 5622 | 1.26 | 6.14E+05 | 9.78E+04 | 5.92E+04 |
| *Tyrannosaurus* | MOR 555 | 6072 | 1.31 | 6.90E+05 | 1.04E+05 | 6.31E+04 |
| *Acrocanthosaurus* | NCSM 14345 | 6177 | 1.45 | 7.74E+05 | 1.05E+05 | 6.40E+04 |
| *Giganotosaurus* | MUCPv-CH-1 | 6908 | 1.32 | 7.88E+05 | 1.14E+05 | 7.01E+04 |
| *Tyrannosaurus* | CM 9380 | 6987 | 1.29 | 7.76E+05 | 1.15E+05 | 7.08E+04 |

B)

| Taxon | specimen | total daily (basal + 12 hours walking) | | | | 18 km + daily BMR | | | |
|---|---|---|---|---|---|---|---|---|---|
| | | BMR [45] | % diff. | BMR [46] | % diff. | BMR [45] | % diff. | BMR [46] | % diff. |
| Tyrannosaurus | BMRP 2002.4.1 | 1.19E+05 | 32.5 | 1.09E+05 | 35.5 | 4.09E+04 | 19.8 | 3.08E+04 | 26.2 |
| *Ceratosaurus* | USNM 4735 | 1.58E+05 | x | 1.48E+05 | x | 4.94E+04 | x | 3.91E+04 | x |
| *Gorgosaurus* | AMNH 5664 | 1.32E+05 | 20.4 | 1.21E+05 | 22.1 | 4.40E+04 | 12.7 | 3.37E+04 | 16.6 |
| *Sinraptor* | ZDM 0024 | 3.81E+05 | x | 3.59E+05 | x | 1.21E+06 | x | 9.78E+04 | x |
| *Gorgosaurus* | AMNH 5458 | 3.60E+05 | 6.2 | 3.37E+05 | 6.7 | 1.17E+05 | 4.0 | 9.37E+04 | 5.0 |
| *Acrocanthosaurus* | NCSM 14345 | 7.81E+05 | x | 7.43E+05 | x | 2.39E+05 | x | 2.01E+05 | x |
| *Tyrannosaurus* | BHI 3033 | 7.12E+05 | 10.0 | 6.73E+05 | 10.6 | 2.26E+05 | 6.6 | 1.87E+05 | 8.0 |
| *Tyrannosaurus* | MOR 555 | 7.93E+05 | x | 7.53E+05 | x | 2.47E+05 | x | 2.07E+05 | x |
| *Acrocanthosaurus* | NCSM 14345 | 8.78E+05 | 9.5 | 8.38E+05 | 10.0 | 2.66E+05 | 6.6 | 2.25E+05 | 7.8 |
| *Giganotosaurus* | MUCPv-CH-1 | 9.02E+05 | x | 8.58E+05 | x | 2.78E+05 | x | 2.34E+05 | x |
| *Tyrannosaurus* | CM 9380 | 8.91E+05 | 1.3 | 847165 | 1.4 | 276543 | 0.9 | 232518 | 1.0 |

A) Cost of transport during daily foraging during and energy expenditure calculated using basal metabolic rate (BMR) estimates per [45, 46] in Kj. B) Comparison of daily energy expenditure (foraging + BMR) between Tyrannosauridae and similar sized basal large bodied theropods.

weaker (Distal limb index = -0.41125+/- 0. 064192 *log SVL+3.0372 +/- 0.17972, $r^2$ = 0.39, p (uncorr)>0.001, n = 77) which may explain why, without correcting for mass, it shows a significant relationship to top speed.

It is clear that when body mass is taken into account, there is an upper limit on running speed that becomes more influential on the life history and ecology of theropods as one approach's ~1000 kgs (Fig 5). This pattern fits with what is expected theoretically [59, 60] and shown through empirical studies [7, 20, 61]. In the largest mammalian land animal of today, white rhinos and elephants, top recorded speeds are much lower than those of smaller animals with shorter absolute limb lengths [7, 61] and their limbs are suspected to be adapted more for reducing locomotive costs at these sizes than fast running [61]. This is not to say that running is not possible for taxa of greater than 1000kg, as white rhinos are recorded at speeds exceeding 11m/s for a Froude value of around 11 (though this is speed record is suspect due to the uncertainty in using car chase speed reports [61]) and elephants at over 6m/s, giving the later a maximum Froude value of less than 5 [62, 63]. Both of these values are comparable to what we estimate here based on [7] for similar sized theropods.

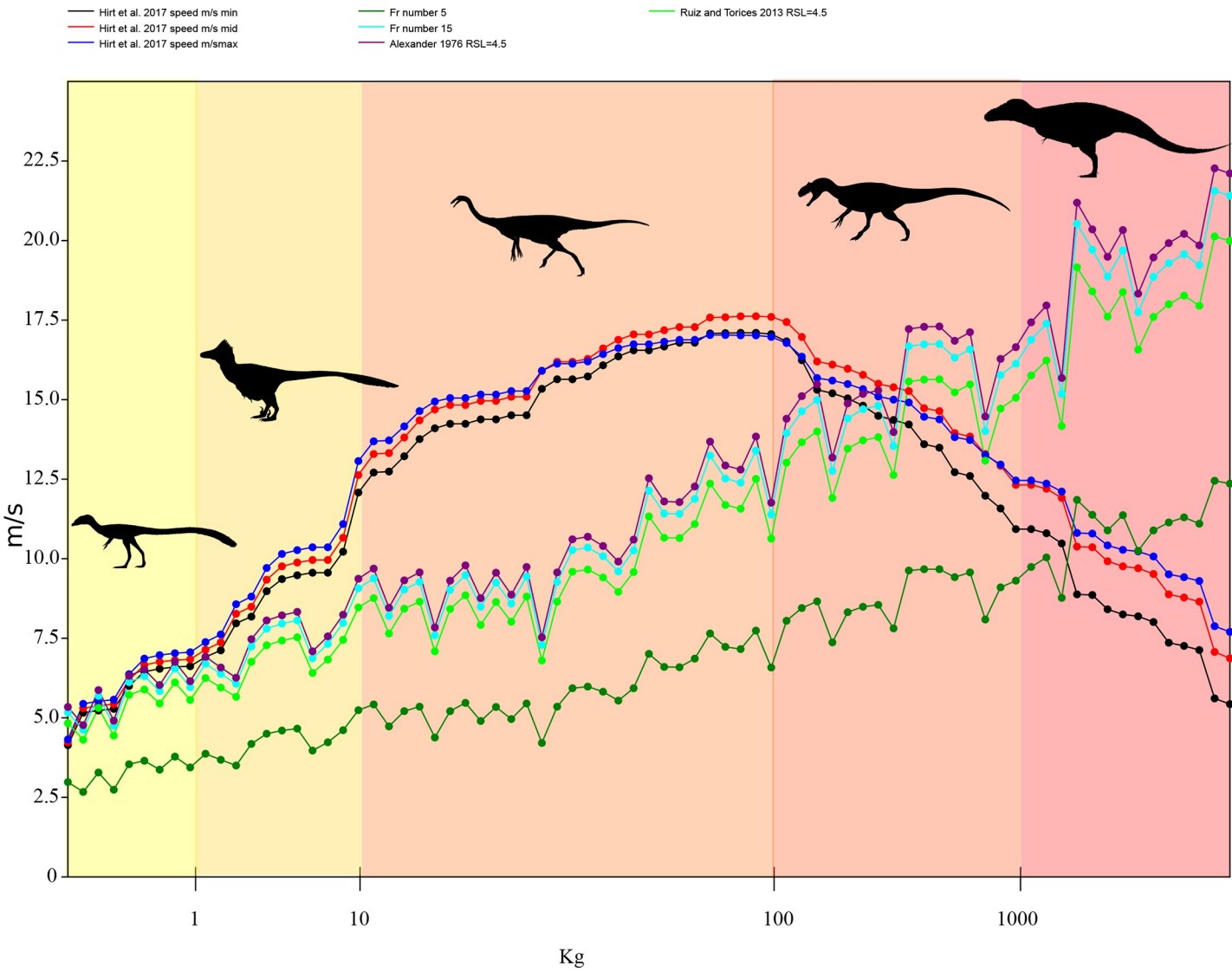

**Fig 5. Pattern of maximum running speed across non-avian theropods.** Observing the effect of increasing body mass on top speed in non-avian theropods by evaluating the difference between various reconstructive methods, including upper and lower confidence intervals for [7]. Note that at smaller body size, less than 100 kg, there is a large and increasing gap between the top speed limit imposed by [7] and top end estimates from other methods. This gap becomes largest in specimens between 10–100 kg indicating that perhaps these specimens had the highest ceiling to increase running speed by exaggerating hindlimb muscle size, altering insertion location, moment arm length, total leg length or stride frequency. Silhouette modified from those in Phylopic image repository (Phylopic.org) created by Joh Conway, Scott Hartman, Emily Willoughby and Matt Martynuik.

While this size class, greater than 1000 kg, only represent a fraction of theropod diversity [64] it represents crucial mid to top level carnivores for much of the Mesozoic since the Early Jurassic [1, 15, 65]. This raises the questions of why certain groups, most notably the tyranno-saurids, elongate their hindlimbs relative to more basal taxa when this costly addition, in terms of growth, was not aiding in increasing speed as they had already maxed out their potential for that. We suggest as one possibility that the likely selective pressure driving this was related to increasing foraging ability or home range size by decreasing the energy spent during low speed locomotion over long distance, as seen in extant taxa [20, 61]. Alternatively (and not mutually exclusive), they may have simply retained this limb proportions from smaller-bodied ancestors

or earlier ontogenetic stages in which these proportions were adaptively significance in terms of increased speed [66, 67].

In many modern hunters, active searching for food does not occupy the entirety of their day [21, 22], though this does increase markedly in scavengers [45]. Of the time spent actively foraging only a fraction of that is accounted for by high speed pursuit. For example, in African wild dogs less than 8% of total hunt distance is traveled at high, yet not top, speed [68] and similar pattern seen in the amount of running stalking time seen in lions [69]. It is probable therefore that larger theropods would likely have not been spent in the active pursuit of prey at top speeds, regardless of the time spent foraging. Furthermore, the total energetic cost of hunting prey (pursuit, capture and killing) in modern larger carnivores is notably higher compared to those who favour small prey [21]. We therefore infer that amongst theropods weighing over 1000kg, given the limitations in top speed performance regardless of limb length due to body-size [7], selection for energetic efficiency was likely significant regardless of the intralimb proportions.

For smaller (<1000 kg) theropods the opposite conditions apply. Not only are they more likely to be small prey specialist, where pursuits are short and prey easy to subdue, limiting the energy losses during hunting. Yet just as importantly these organisms are themselves potential prey items to larger theropods. This means that they have a strong selective pressure to obtain high top speed, especially with a short acceleration time, to facilitate escape. Including the possibility of a more crouched stance in the smallest theropods (<40kg) only exacerbates the difference between body mass derived limitations and the maximumly speed possible from limb elongation. Thus, we find two opposing selective pressures across theropod hindlimb, one at small size to maximize speed which decreases as you get larger to focus more on energetic savings in mid-sized to large members of the theropod community.

**Why tyrannosaurids?.** In looking for the origins behind the trend of long leggedness in tyrannosaurids, and the potential ecological and behavioural underpinnings for it, one must first determine if it a plesiomorphic feature of a wider Tyrannosauroidea or even coelurosaurian condition. That the coelurosaur condition is characterized by elongated hindlimbs is unlikely as other basal coelurosaurs such as compsognathids show reduced hindlimbs, among the lower third of the dataset and though the tibia is incomplete *Zuolong* shows values closer to *Deinonychus* (59th) than *Tanycolagreus*. While some basal tyrannosauroids do show elongated hindlimbs compared to femoral circumference (S4 and S5 Tables) such as *Guanlong* (1st), *Tanycolagreus* (12th), and *Moros* (11th), others such as the basal *Coelurus* (65th) or *Dryptosaurus* (42nd), the latter of the two is larger (> 1000kg) and closer to tyrannosaurines, ranks in the lower half. Additionally, if one were to reconstruct the femoral circumference of *Dilong* from its femoral width it would rank in the bottom quartile at around 64th. This combined with significant uncertainty due to the number of partial specimens at the base of the tyrannosauroid tree as well as the potential for the Megaraptora to be basal tyrannosauroids [70], paints an uncertain picture of how to reconstruct the evolution of hindlimb elongation in this clade. What we can say is that all the small long-legged basal members of this clade are well below the inflection point of selective pressures for speed versus efficiency. As such their position as mid-level predators in their ecosystem who were potential prey themselves could lead to species specific selection pressures being confused with clade wide trends. Finally, as we do not have a good understanding of the size of basal members of Tetanurae or Orinoides, though we suspect they were significantly smaller our cross over point [2, 15, 71]. Without these fossils, we cannot assess if derived tyrannosaurs retained the elongated hindlimbs of their small ancestors as they evolved gigantism or if this was a secondary elongation event confined to the later members of Eutyrannosauria.

Despite this the fact that both subadult and mature allosauroids, tyrannosaurids, and other tetanurans were too large to access upper range of speed due to their hindlimb length, raises the question of why they differ so much in relative limb length. One potential explanation is difference in prey choice. Sauropods were rare in those communities where Tyrannosauridae existed [72], with only a single taxon, *Alamosaurus*, known from North America restricted to the Southernmost part of *Tyrannosaurus*' range [73], and two (c.f. [74]) small sauropods from the Nemegt which are minor members of the fauna [75]. For tyrannosaurids the most common larger prey taxa are herds of ceratopsians and hadrosaurs which are on the order of 1/5-1/10 the mass of the sauropod prey available to the larger allosauroids and basal tetanurans [64]. Furthermore, sauropods were a ubiquitous component of the ecosystems of these more basal large theropods [76], presenting a common and calorie dense meal source either through direct predation or carcass scavenging. While it is likely that much of the prey captured for theropods were juvenile and subadult specimens regardless of the prey species [77], sauropods would still provide a much larger meal with many species with over 40% of the population consisted of individuals of 3500kg or more [76]. In addition sauropod trackways indicate they tended to walk at slow speeds [78], and their size alone strongly suggests they would have a limited top speed significantly below that of their contemporaneous larger theropod faunas [7]. Thus, sauropods would provide an abundance of larger, slower and more energy dense food resources for more basal large theropod clades. Conversely we are suggesting that the pressure for obtaining more kills due to the fact that each kill provides less resources, thus necessitating minimizing energy expenditure per hunt and maximizing resource extraction per kill, especially if that kill is shared amongst a group, influenced selection for longer limbs in Tyrannosauridae.

Hunting the relatively smaller and faster hadrosaurs and ceratopsians may also have been facilitated by group behavior in tyrannosaurids, something previously documented by track and body fossils in large theropods [47, 79]. Juveniles, less than 10–15 years of age [80, 81] would still be in the zone where their long legs could be used to maximize top speed, potentially helping run down faster prey items. Beyond this it has been shown that amongst pack hunting animals employing strategy or communication between individuals can allow them to capture prey that is faster than any one individual [82]. Combining these factors we find that pack hunting would only increase the energetic savings differential even more dramatic between tyrannosaurs compared to allosauroids. For example, if we assume a tyrannosaur "pack" consisting of two adults around the size of BHI 3033 or MOR 555 and two subadults with femora approaching 1 m in length and 2500kg in mass and two juveniles the same size as "Jane" the savings versus a similar sized and demographically distributed group of *Acrocanthosaurus* or *Saurophaganax* is between 4000-4300kg worth of prey. This corresponds to about the mass of a 1–2 hadrosaurids [44] or 28–30 days of total energetics for the groups. If similar to modern large terrestrial carnivores the majority of hunts end in failure with only a 20–30% success rate [21], such a savings would reduce the necessity for multiple hunts, where during each on beyond the loss of energy in a failed capture this there is the inherent risk from injury either during the pursuit or capture itself. Such a large amounts of savings, corresponding to several large kills per year, would have significant effect on survivorship of the group.

Finally, there is the fact that meat acquisition does not necessarily have to exclusively come from the capture and killing of live prey items. Most modern primary predators and, likely, extinct ones such as large theropods, probably incorporate a significant fraction of carrion into their diet [45]. We know of several occurrences of likely scavenged tyrannosaurid feeding traces [77, 83, 84] indicating some facultative carrion usage did occur. Recent work [45] has estimated that scavenging would have been most important to mid to large, but not extremely large, sized theropods around the range that we find mass induced upper limits on top speed.

While we may not agree with the assumptions and assertions of the level of scavenging suggested by [45], we do suggest that this is another line of evidence of the increasing role of energy efficiency over long distances locomotion. Given the data we have presented here saving multiple days' worth of feeding requirements due to reduced energetic demands by increased leg length in large tyrannosaurids. Any adaptation that helps reduce the costly and potentially hazardous search, capture, killing and defending a kill would be a significant evolutionary advantage for that lineage and may have been one of the keys to their success in the Late Cretaceous c.f. [85].

## Conclusions

We find that traditional, simple metrics, notably the distal limb index, fail to reflect true measures of cursorial and especially top speed potential across Mesozoic theropods. When direct comparisons of similar sized individuals are performed, several clades—most notably the compsognathids and basal birds which show high levels of distal limb elongation, do not show comparable total limb relative lengths or top speed to microraptorines or basal troodontids. Without accounting for the allometric influence on any of these limb metrics we remain highly skeptical of their broad application. Additionally, we also show that when we include the fact that there is a parabolic distribution of top speeds, with local maxima between 500-2000kg depending on the Froude number used to estimate speed, there is no significant relationship between distal limb index (or indeed any other commonly used hindlimb index) and top speed across theropods. We argue that selection for intralimb lengths is likely multifaceted, clade specific and unlikely to be captured in a simply, overarching metric.

Factors such as clade history, diet and prey capture methods, for example the role of the hindlimb in subduing prey in eudromaeosaurs [86] likely has implication for why they tend to have short metatarsals, all combine with speed and cost of transport influences to shape the final product. Despite saying this we do propose that, at a first order of magnitude, we can argue that their body size likely has a major role. Body sizes is here postulated to be strongly influential in the shifting the speed versus endurance/ energy savings balance in the paleobiology of theropods. Smaller taxa are more likely to take smaller prey, which reduces foraging and capture costs but conversely are they themselves much more likely to predated upon. For them a fleet foot may be the difference not just in a full or empty belly but in life or death. In larger taxa this balance shifts to be more "waste not, want not" as they are much less likely to be hunted while they are searching for prey.

We also find that amongst the large bodied theropods tyrannosaurids show markedly reduced values of cost of transport due to their elongated limbs. While their body size makes this unlikely to be of much value in increasing running speed, it does significantly save on the cost of daily foraging expenditures. These savings, up to several tones of meat per year per individual, would dramatically reduce the need to engage in the costly, dangerous and time-consuming act of hunting. One thing to keep in mind is that the cost of transport budget are cumulative. Thus even if it is difficult to calculate exact values for daily energy expenditure for a taxon, the accruing of substantial savings per step every day of an organisms existence would results in significant and impactful differences in the energy budget regardless of which method is used to estimate caloric requirements. As top speed is both not significantly different amongst similar sized large bodied theropods, regardless of limb length, nor is it considered a trait that is commonly optimized for during selection due to its rare use [61], these small continually additive savings in CoT are much more impactful for an organism survivorship. In addition other factors, such as hunting strategy, environmental conditions, prey

abundance and difficulty in subduing it all likely have a much larger role than upper speed limits in shaping the ecological role of these mid to top level predators.

When coupled with the evidence that tyrannosaurids were, at least on occasion, living in groups as well as the fact their primary prey was on average smaller and more elusive than the sauropods that were a major component of the diet of more basal large theropods, this paints a picture where efficiency would be a major evolutionary advantage. Reducing the energy spent locating, pursing and subduing prey has a myriad of benefits and allows both for a reduction in number of kills needed to sustain a set number of animals but also allows them to devote excess resources to other life history aspects. While we cannot clearly ascertain if the "legginess" of tyrannosaurs was an adaptation itself or the retention of the ancestral condition of elongated hindlimbs as gigantism evolved in this clade, both options present interesting evolutionary scenarios with broader implications for the paleobiology and paleoecology of the Late Jurassic to Late Cretaceous ecosystems of Laurasia. Future work in this area will help elucidate which path this extremely successful clade took as they replaced carcharodontosaurs as the apex predator of the Upper Cretaceous [87].

Interestingly, additional analyses support the hypothesis that tyrannosaurids were more agile (that is, capable of turning more rapidly and with a smaller turning radius) than other comparable-sized large-bodied theropods [9]. This similarly reflects a specialization with Tyrannosauridae for hunting large-bodied ornithischians such as hadrosaurids and ceratopsids themselves likely more mobile and agile than sauropods. When combined these two lines of evidence for an energy efficient, yet still nimble, design of the Tyrannosauridae hindlimb reflect a likely long-distance stalker with a final burst to the kill likely in a pack or family unit, similar to modern wolves. This further reinforces the notion, that beyond being the apex predator of the latest Cretaceous Laurasian ecosystems, the tyrannosaurids were amongst the most accomplished hunters amongst large bodied theropods. We find that their anatomy, at once efficient and elegant, yet also capable of burst of incredible violence and brute force, lives up to their monikers as the tyrant kings and queens, of the dinosaurs.

## Supporting information

**S1 Table. Snout to vent (SVL) dataset.** Includes measurement data, hindlimb indices and regressions for 93 specimens for 71 different genera of avian and non-avian theropod. SVL = snout to vent length in mm, F = femur length in mm, T = tibia length in mm, MT = maximal metatarsal length in mm, HL = hindlimb (F+T +Mt) length in mm.
(XLSX)

**S2 Table. Running speed estimates from hindlimb lengths for SVL dataset.** Note that the transition from walking to running occurs around Fr = 0.5[10]. Fr = Froude number, Alexander = [31], Ruiz and Torres = [88].
(XLSX)

**S3 Table. Comparison of running speed from hindlimb lengths to body mass limitations based on [7] for subset of SVL dataset.** Postural scaling based on based on extant ground birds from data in [40]. FC = femoral circumference in mm. Mass = body mass in kg, Leginess = HL/ mass^1/3, MT. mass ^1/3 = Mt/mass^1/3.
(XLSX)

**S4 Table. Comparison of running speed from hindlimb lengths to body mass limitations based on [7] for hindlimb only dataset.**
(XLSX)

**S5 Table. Comparison of femoral circumference vs hindlimb length for hindlimb only dataset.**
(XLSX)

**S6 Table.**
(XLSX)

**S1 Fig. Comparison of maximum running speed vs CoT for selected tyrannosaurs vs other large theropods.** For all analyses green shaded box plots are basal theropods, red is tyrannosaurs. A) Smaller bodied specimens, mass range between 660-688kg, Ceratosaurs (USNM 4735) vs juvenile Tyrannosaurus rex (BMRP 2002.4.1) and Gorgosaurus (AMNH 5664). B) Midsized specimens, mass range between 2375-2430kg, Sinraptor (ZDM 0024) vs. Gorgosaurus (NMC 2120, AMNH 5458). For data see S6 Table.
(PDF)

**S2 Fig. Comparison of maximum running speed vs CoT for selected tyrannosaurs vs other large theropods continued.** For all analyses green shaded box plots are basal theropods, red is tyrannosaurs. C) Large specimens, mass range between 6070-6170kg, Acrocanthosaurus (NCSM 14345) vs adult Tyrannosaurus rex (MOR 555) D) Largest specimens, mass range between 6900-7000kg, Giganotosaurus (MUCPv-CH-1) vs. adult Tyrannosaurus rex (CM 9380, AMNH 5027). For data see S6 Table.
(PDF)

## Author Contributions

**Conceptualization:** T. Alexander Dececchi.

**Formal analysis:** T. Alexander Dececchi.

**Investigation:** T. Alexander Dececchi, Hans C. E. Larsson.

**Methodology:** T. Alexander Dececchi.

**Visualization:** T. Alexander Dececchi, Hans C. E. Larsson.

**Writing – original draft:** T. Alexander Dececchi.

**Writing – review & editing:** T. Alexander Dececchi, Aleksandra M. Mloszewska, Thomas R. Holtz, Jr., Michael B. Habib, Hans C. E. Larsson.

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
