## [Decision Letter · Decision Letter 0]

29 Oct 2019

PONE-D-19-26887

The fast and the frugal: Divergent locomotory strategies drive limb lengthening in theropod dinosaurs

PLOS ONE

Dear Dr. Dececchi,

Thank you for submitting your manuscript to PLOS ONE. After careful consideration, we feel that it has merit but does not fully meet PLOS ONE’s publication criteria as it currently stands. Therefore, we invite you to submit a revised version of the manuscript that addresses the points raised during the review process.

I must apologise for the fact this has been on my desk for a week. The reviewers had quite extreme differences of opinion with regards to the manuscript so I have been through it myself. Whilst reviewer 1 suggests rejecting the manuscript, I am hoping that their comments and those of reviewer 2 will help shape this manuscript into something publishable in the future.

There is a lot of work to be done to make the manuscript publishable from its current form. This includes:

Responding to both reviewers comments. In particular note reviewer 1's comments about formalising hypotheses, and framing the overall focus of the manuscript.Your methods section needs a lot of work explaining certain decisions, e.g. crouch amount (see reviewer 1 comments). Any models you use, please explain in full in the methods, and discuss later their limitations (PLOS doesn't have a page limit, so do not skimp on making things clear).Tightening up the language throughout the manuscript (there are many grammatical errors throughout, beyond those highlighted by reviewer 1 and 2). Please add line numbers to any resubmitted copy to allow for easier checking/commenting.Check your formatting throughout. Alignment within tables, numbers of significant figures (e.g. do we really need 9 significant figures in cost of foraging/year?), superscripting etc.

Please note there is a "due date" automatically assigned below, but take as long as you need to tidy and revise this manuscript.

We would appreciate receiving your revised manuscript by Dec 13 2019 11:59PM. To enhance the reproducibility of your results, we recommend that if applicable you deposit your laboratory protocols in protocols.io, where a protocol can be assigned its own identifier (DOI) such that it can be cited independently in the future. For instructions see: http://journals.plos.org/plosone/s/submission-guidelines#loc-laboratory-protocols

We look forward to receiving your revised manuscript.

Kind regards,

Andrew Cuff

Academic Editor

PLOS ONE

Journal Requirements:

2. Please ensure that you refer to Figure 5 in your text as, if accepted, production will need this reference to link the reader to the figure.

Reviewers' comments:

Reviewer's Responses to Questions

**Comments to the Author**

1. Is the manuscript technically sound, and do the data support the conclusions?

Reviewer #1: No

Reviewer #2: Yes

2. Has the statistical analysis been performed appropriately and rigorously? 

Reviewer #1: No

Reviewer #2: Yes

3. Have the authors made all data underlying the findings in their manuscript fully available?

Reviewer #1: Yes

Reviewer #2: Yes

4. Is the manuscript presented in an intelligible fashion and written in standard English?

Reviewer #1: No

Reviewer #2: Yes

5. Review Comments to the Author

Reviewer #1: In the present study, the authors predict running speed and cost of transport in theropod dinosaurs. Unfortunately the study is not executed well. The manuscript is poorly constructed, and the narrative is extremely difficult to follow in many places. The grammar and legibility require considerable improvement, and the authors' non-scientific language in the conclusions is inappropriate. The justification for conducting the study is not entirely clear, and there does not seem to be any obvious hypotheses or research questions. The methodology requires the authors to bring in multiple numerical models from existing publications. However many of these predictive models are not explicitly laid out in the text, and the numerous assumptions associated with the models are not discussed. Holding posture as constant, for example, is a massive assumption given the considerable literature studying the postural shift occurring in the theropod lineage (Allen et al, 2013, Nature). Ultimately I was not sure what I was supposed to take away from the study. Is this a methodological study attempting to present a new metric for estimating running speed in fossils? Or rebut an existing one? Or is this a comparative functional morphology study looking at the evolution of running speed and cost of transport in the theropod lineage? If the purpose of the study was to rebut the Person and Currie 'distal hind limb index' by incorporating the parabolic model of Hirt et al, that would be a fine endeavour and would be achievable in 1500 words as a comment or reply to the original study. Unfortunately, in its present form, I do not believe this study is a meaningful contribution to the field of dinosaur palaeobiology.

Reviewer #2: This paper examiness limb proportions and relates them to estimates of speed and cost of transport / foraging costs in non-avian theropod dinosaurs. They argue that limb proportions are not generally reliable for predicting speed (in large part because the equations do not take into account the effect of scaling at large body sizes), and that the long legs of large-bodied tyrannosaurids lowered the energetic cost of foraging compared to other large-bodied theropods.

I agree with the overall conclusions of the article. Limb proportions cannot be sufficient for speed estimates given the restrictions faced by large taxa. I'm not sure that many paleontologists disagree on this point these days, and I'm not sure that other researchers use 'cursoriality' to only mean top speed. It makes sense that proportionately long legs would make for better cost of transport, particularly given that, as far as I can tell, the equation being used is derived form Pontzer 2007 that only uses hip height.

I think the main weakness of the paper is that several of the fundamental equations are based on samples across animals that find overall trends based on log-transformed data. I don't disagree with these papers' conclusions- there is a relationship between size and absolute speed, and metabolic cost and hip height. However, using equations based on these big differences to examine fine-grained within clade differences is less reliable than more specific models. For instance, the Y scale in figure 3 shows that estimates from the mass limitation paper can be up to about double those based on Froude numbers. This combined with other factors, such as log-transformed data tending to produce visually tight clusters of disparate data at large values, and the lack of confidence interval/errors for some of these estimates, can produce precise-looking estimates and trends where it is unclear how confident we should be in these estimates. There may not be an applicable fix currently for this issue, but I do think that it might be worth reminding readers that even though regressions produce point estimates, that there is some confidence interval around that estimate that might blur otherwise clear-looking patterns.

I think that any paper that deals with cursoriality or speed and discusses theropods should include at least a brief discussion concerning gait and grounded running. Unfortunately, there is confusion in the literature about what "running" means even in papers dealing specifically with top speeds, and so it's best not to contribute to this confusion. Any equations derived from mammal data concerning speeds should also be defended given theropods ability to achieve running dynamics while maintaining dual support.

More detailed comments follow:

Line numbers should always be included in manuscripts.

The abstract seems a bit strong in regards to the impact of mass. Is body size really often overlooked when discussing speed in theropods? A few holdouts occasionally opine about Tyrannosaurus galloping but the serious literature hasn't bother with those sorts of ideas for a long while it seems to me. Also is cursoriality the equivalent of speed? It's not clear to me that Persons and Currie mean speed.

The variables in the supplemental tables should be described and explained somewhere.

Throughout the manuscript interlimb is used when I think intralimb is meant. Given that the paper partially seems to be intended as a response to Persons and Currie, it is disheartening that Persons name is consistently misspelled as Parson throughout the text.

Figure 2- B is almost entirely redundant with A. Replot as one panel with Halszkaraptor receiving some unique symbol so that it can be picked out.

"sharp demarcation" p15. The point estimates seems relatively distinct but there's no error on them. How much messier does it look when different estimates for each taxon are plotted? The next sentence reveals that estimates for one taxon can vary almost 3m/s.

Table 3: B is confusing, needs to be explained or laid out better.

Is Figure 5 referenced in the text anywhere?

I have made a few more notes in the attached pdf and have highlighted some obvious grammar errors.

6. PLOS authors have the option to publish the peer review history of their article (what does this mean?). If published, this will include your full peer review and any attached files.

Reviewer #1: No

Reviewer #2: No

---

## [Author Response · Author response to Decision Letter 0]

9 Dec 2019

Response to reviewers' comments for manuscript PONE-D-19-26887:

“The fast and the frugal: Divergent locomotory strategies drive limb lengthening in theropod dinosaurs”

Reviewer # 1 (Comments to the Author)

In the present study, the authors predict running speed and cost of transport in theropod dinosaurs. Unfortunately the study is not executed well. The manuscript is poorly constructed, and the narrative is extremely difficult to follow in many places. The grammar and legibility require considerable improvement, and the authors' non-scientific language in the conclusions is inappropriate. The justification for conducting the study is not entirely clear, and there does not seem to be any obvious hypotheses or research questions. The methodology requires the authors to bring in multiple numerical models from existing publications. However many of these predictive models are not explicitly laid out in the text, and the numerous assumptions associated with the models are not discussed. Holding posture as constant, for example, is a massive assumption given the considerable literature studying the postural shift occurring in the theropod lineage (Allen et al, 2013, Nature). Ultimately I was not sure what I was supposed to take away from the study. Is this a methodological study attempting to present a new metric for estimating running speed in fossils? Or rebut an existing one? Or is this a comparative functional morphology study looking at the evolution of running speed and cost of transport in the theropod lineage? If the purpose of the study was to rebut the Person and Currie 'distal hind limb index' by incorporating the parabolic model of Hirt et al, that would be a fine endeavour and would be achievable in 1500 words as a comment or reply to the original study. Unfortunately, in its present form, I do not believe this study is a meaningful contribution to the field of dinosaur palaeobiology.

We thank the reviewer for pointing this out. We have now corrected grammar and legibility throughout the manuscript. We have also modified the language and tone of the specific sections identified by the reviewer. 

We respectfully disagree with the reviewer with regard to narrative. The various aspects of the multifaceted issue highlighted within the manuscript have been identified and addressed in sequence. The presentation sequence that we use, that is: 1. analyzing previous estimators of cursorial ability, 2. including the limiting factor of body size, and 3. evaluating a possible energetics explanation for the largest theropods as they exceed the size limitation of maximum running speed, is a necessary progression to show the audience why simple metrics alone cannot tell the entire ecological and paleobiological picture. This progression was specifically chosen because it highlights that central fact, that allometry and absolute mass have profound effects on paleoecology and energetics, and makes our arguments in the discussion and conclusions as accurate and clear as possible. 

As part of our work, we included a section on examining how various simple metrics reflect these overarching behaviours (i.e., distal limb index of Person and Currie, 2016). This was not the central focus of our research. Nevertheless, our evaluation and rejection of the index proposed by Person and Currie (2016) as an accurate means to estimate highly cursorial and high speed runners amongst non-avian theropods is an important contribution to the overall literature, despite not being the major focus of this study. Its inclusion here is because it and other similar ratios are a central pillar of many studies, functional and phylogenetic , to infer behavioural repertoires . As such we felt if we did not include an analysis section on the reliability of this and similar metrics, which touch upon but is not central to the broader scope of work, our manuscript would be incomplete and less impactful. 

With regard to justification and lack of a clearly stated hypothesis, while our justification is clearly stated in the opening lines of the abstract, and in the introduction in the original (lines 30-31 and 96-97 in the clean text) We have now added a hypothesis statement to the newest version of the manuscript to address the reviewer’s concern (lines 90 -93 in the clean text version). 

Regarding the use of specific models in our study, we rely on modeling and mathematical estimations when performing any reconstruction. This is necessary in order to gain the prerequisite information to enter into our analysis. In this study, reliable, peer reviewed data sources were selected as a starting point and then modified as necessary to fit with the scope of our study. This strategy has been adopted in countless other peer-reviewed publications, including several by each of the main authors, where input data derived from other published sources is used. In these studies, even simple values such as linear measurements, which often differ slightly between left and right sides or between authors taking them, are constantly used as if they were a single fixed variable with no associated errors, and this is not commonly seen as a major flaw by the scientific community. All of these input equations and variables sourced have been appropriately referenced so that readers can refer to the procedures in the specific peer reviewed publications if required. 

Regarding the reviewer’s objection to postural changes based on Allen et al. (2013), we argue that this study’s results do not invalidate or drastically impact our own for several reasons. Our study focuses specifically on running speed relationship with leg length and body mass, as well as how limb elongation alters the long term cost of transport in large theropods. Therefore, our results are not significantly affected by discussions of center of mass shifts within the Neoavian stem. At high speeds, such as what we are testing here since Froude numbers of 5 or greater are a fast run, large animals adopt a more upright stance, as seen in modern mammals at between 100-300kg (Bertram and Biewener, 1990; Biewener, 1990). Although smaller mammals remain proportionally more crouched, the challenge is to determine how significant this shift would be in the Theropods, especially as many small mammals often occupy niche spaces that include behaviours not suspected for non-avian taxa such as burrowing or climbing. We chose our crouching value, 0.8 of total limb length, and set it as a constant. This value was selected as it closely approximated that see in large (10+ kg) modern birds (Gatesy and Biewener 1991; Birn Jeffery et al. 2014; Bishop et al. 2018). While this value does vary somewhat in mid-stance allometrically in modern avians, but in taxa of around 1.0 kg and above this value, when you use just the three limb elements (Femur tibia and metatarsus) we used for non-avian theropods, is greater than 0.6-0.8 times hindlimb length (Gatsey and Biewener 1991; Bir-Jeffery et al. 2014; Bishop et al. 2018) in level running. Given that a) the vast majority of our taxa are above 1.0 kg (18/ 22 in the SVL dataset and 68/77 in the Leg only), and b) that there is a significant shift to a more crouched posture within Neornthines compared to the taxa investigated here (Allen et al. 2013), we defend this value as likely being within the range seen in non-avian theropods running at higher speeds. For our foraging cost calculations, this would be even less of an issue, as all individuals in our comparisons were estimated at minimally 660 kg, well within the range of upright stances using either a mammalian or avian model. 

Beyond this, it is unlikely that the data from Allen et al. (2013) would significantly alter our analysis results as Allen et al. (2013) state that the origin of the “crouched’ posture occurs at Tetanurae which, in their phylogeny, is represented by Dilophosaurus and includes all taxa more crownward than the coleophysoids. In contrast, our analysis has only 5 (out of 93) members of the SVL dataset, and 5 (out of 77) in our ‘hindlimb only’ dataset that are more basal than this node. Thus, shifts in COM and posture occurring here should have little effect on the majority of our data. Furthermore, in the supporting information Figure 1 in Allen et al. (2013), they show very little dorsoventral shifting in COM observed between Theropoda and Ornithothoraces (nodes 5-14 in their phylogeny), and any significant alterations happens well beyond our taxonomic sampling. We feel that shifts in this axis, rather than cranial caudally, is much more important in determining hip height, especially during high speed running as examined here.

In addition, there are several other factors that make it unlikely that the COM shifts seen in Allen et al. (2013) are highly influential on the results presented here. First, we have some issues with the data Allen et al. (2013) used, including the fact that their cf. caenagnathus specimen (CM78001) listed by Allen et al. with a mass of between 9-16 kg, a pelvic limb of 0.539 m, and a femur length of 0.17m. This specimen is actually the type of Anzu, a fairly large oviraptorosaur. with a femur measuring 0.505 m, hindlimb length of around 1.2+ m, and a mass estimated at 200-300kg (Lamanna et al. 2014). Because of this discrepancy we suggest this data point is likely suspect, and as it comes directly before Eumaniraptora (and is the only species recorded for the node Maniraptora), that makes suggestions of any shifts that occur between Maniraptora and Eumaniraptora more uncertain. There are other issues, such as the mass estimate for Archaeopteryx being between 0.066-0.13kg while other volumetric methods have it well above 0.2 kg (Yalden 1983; Elzonwski 2002; Pontzer et al. 2009 amongst others), which effects the proposed shift amongst the basal most avians as well. This inaccuracy becomes important because, as stated in the supporting information for Allen et al. (2013), relative COM is related to body size. Proportional Caudofemoralis size also scales allometrically. This helps exaggerate any observed shift (especially if one removes the erroneously reconstructed cf. caenagnathus) from the basal maniraptoriform Struthiomimus between 394-741 kg to the eumaniraptorans Velociraptor at 9.9-18.3 kg. Of note, Microraptor, whose COM would be critical for understanding if this change is purely allometric or has some function significance amongst paravians, does not have a reconstructed caudofemoralis mass. Therefore, no discussion can be made on its behalf. One wonders whether any signals for a significantly different posture amongst derived maniraptorans compared to more basal tetanurans is mostly due to the size of the specimens selected. Allen et al. (2013) also does not include any complete small bodied more basal coelurosaurs such as Compsognathus or Sinosauropteyrx, thus any discussion of how stance amongst small bodied taxa differed , as well as any challenge to our use of metrics to reconstruct hip height and speed, is not possible. 

 Finally, as small bodied paravians possessed elongated leg feathers that would, if held at a lower angle such as a deep crouch, become damaged on the substrate during locomotion, this argues against them adopting the level of crouching seen in similar sized modern birds. We do have trackways from individuals likely well below 0.5kg in mass (Kim et al. 2018) without evidence of feather drags. While this does not prove that they must have had a more upright posture, it is suggestive that these taxa had proportionally higher stances and hip heights even in small bodied taxa. And even if we assume a crouched position during the run similar to modern birds of ~1 kg for our smaller bodied specimens, this would only reinforce our finding that hindlimb length is the limiting factor for maximum burst speed from them, as it would have further lowered these values below the level estimated using Hirt et al. (2017) mass based equation. This would only strengthen our argument for the selective value of increasing leg length amongst small (not large) theropods in order to become faster to avoid predators and secure prey. 

Despite these arguments, we have included a new paragraph and analyses incorporated into Supporting Tables 3 and 4 that adjusts the posture of our specimens based on a regression from modern ground birds from the data of Gatesy and Biewener (1991). We would like to point out that this change offers only minor alteration to specimens below 40kg in body mass if we use their data. This includes the foot as part of the hindlimb (something we did not do for non-avian theropods), at which point the value shifts to 0.8 and thus we maintained that constant for all larger bodied specimens. More recent work by Bishop et al. (2018) using just the three leg elements we did to constitute “leg length” shows even less prevalent crouching levels in modern ground birds, with no specimen at or above 0.6kg having hip levels less than 80% of leg length. While there is an allometric effect it would suggest that, if anything, we are over estimating the level of crouch in larger taxa, something that would again only hasten the intersection point where body size precludes longer limbs having a positive effect on speed. That said, we did perform an adjustment in addition to our original data to incorporate crouching and this adjustment, while it does lower the absolute maximum speed attained, does little to the relative differences between individual specimens or clades discussed. Nor does it alter the overarching finding of a shift from a drive to maximize running potential in small to medium sized theropods, while reducing transportation costs dominate at larger size classes. 

In a more general response to this reviewer, we believe that a paper combining the testing of existing estimators for cursoriality with evaluation of the major selective pressures driving increased leg length amongst different theropods clades, and evaluating the energetics of foraging amongst large bodied theropods, does make a substantial contribution to the paleobiological literature. The fact that we decided to combine several different, but related, analyses and datasets in a single manuscript is to us not a drawback, but a strength. We would therefore like to thank the reviewer for their comments, and would be keen to work with them on specific issues and sections that they deem as requiring more work. However, with respect, we do not see the foundation of the reviewer’s argument for rejection as being solid or providing any useful path forward in addressing this problem. 

Reviewer #2: This paper examiness limb proportions and relates them to estimates of speed and cost of transport / foraging costs in non-avian theropod dinosaurs. They argue that limb proportions are not generally reliable for predicting speed (in large part because the equations do not take into account the effect of scaling at large body sizes), and that the long legs of large-bodied tyrannosaurids lowered the energetic cost of foraging compared to other large-bodied theropods.

I agree with the overall conclusions of the article. Limb proportions cannot be sufficient for speed estimates given the restrictions faced by large taxa. I'm not sure that many paleontologists disagree on this point these days, and I'm not sure that other researchers use 'cursoriality' to only mean top speed. It makes sense that proportionately long legs would make for better cost of transport, particularly given that, as far as I can tell, the equation being used is derived form Pontzer 2007 that only uses hip height.

I think the main weakness of the paper is that several of the fundamental equations are based on samples across animals that find overall trends based on log-transformed data. I don't disagree with these papers' conclusions- there is a relationship between size and absolute speed, and metabolic cost and hip height. However, using equations based on these big differences to examine fine-grained within clade differences is less reliable than more specific models. For instance, the Y scale in figure 3 shows that estimates from the mass limitation paper can be up to about double those based on Froude numbers. This combined with other factors, such as log-transformed data tending to produce visually tight clusters of disparate data at large values, and the lack of confidence interval/errors for some of these estimates, can produce precise-looking estimates and trends where it is unclear how confident we should be in these estimates. There may not be an applicable fix currently for this issue, but I do think that it might be worth reminding readers that even though regressions produce point estimates, that there is some confidence interval around that estimate that might blur otherwise clear-looking patterns.

I think that any paper that deals with cursoriality or speed and discusses theropods should include at least a brief discussion concerning gait and grounded running. Unfortunately, there is confusion in the literature about what "running" means even in papers dealing specifically with top speeds, and so it's best not to contribute to this confusion. Any equations derived from mammal data concerning speeds should also be defended given theropods ability to achieve running dynamics while maintaining dual support.

More detailed comments follow:

Line numbers should always be included in manuscripts.

We thank the reviewer for point this out, and have now added line numbers to the revised version. 

The abstract seems a bit strong in regards to the impact of mass. Is body size really often overlooked when discussing speed in theropods? A few holdouts occasionally opine about Tyrannosaurus galloping but the serious literature hasn't bother with those sorts of ideas for a long while it seems to me. Also is cursoriality the equivalent of speed? It's not clear to me that Persons and Currie mean speed.

Re-reading Persons and Currie’s text, it is clear that they mean to evaluate running ability, and repeatedly reference maximum running speed, while noting that they really can’t test that using their proposed metrics. In addition, having listened to several talks by other members of this research group which all have used CLP score as a proxy for running speed in their taxon of interest. As for Tyrannosaurus top speed, even many serious works have postulated speed as being a major factor in their extended distal limb segments, even if they don’t think they ran at a gallop. Our finding that speed is highly limited by mass, not just in the largest of the large specimens but likely in all specimens over 1000kg, has a broader impact on our reconstruction hunting behaviour and paleoecology for theropods large, medium and small. 

The variables in the supplemental tables should be described and explained somewhere.

We agree with the reviewer. These have now been listed in the description of Figure 3 as well as in the supporting tables, where they appear.

Throughout the manuscript interlimb is used when I think intralimb is meant. Given that the paper partially seems to be intended as a response to Persons and Currie, it is disheartening that Persons name is consistently misspelled as Parson throughout the text.

We thank the reviewer for pointing this out. We have now corrected this throughout the manuscript.

Figure 2- B is almost entirely redundant with A. Replot as one panel with Halszkaraptor receiving some unique symbol so that it can be picked out.

We agree with this comment, and have made the suggested modification. 

"sharp demarcation" p15. The point estimates seems relatively distinct but there's no error on them. How much messier does it look when different estimates for each taxon are plotted? The next sentence reveals that estimates for one taxon can vary almost 3m/s.

With apologies for the confusion, we would like to clarify that this is the estimated speed across various speed estimator, and not the error. For example the comparison of Changyuraptor compared to Sinosauropteyrx, two species similar in SVL (2% difference) using any individual metric, we find that the former has a 131% higher estimated top speed than the later as show in Supporting Table 2. Thus, while the top speed estimate between metrics for any individual taxon may vary, and these are the different numbers the reviewer alluded to, the differences between individuals under the same estimator remain large. We have now added a sentence in the revised manuscript to clarify this point. 

Table 3: B is confusing, needs to be explained or laid out better.

Is Figure 5 referenced in the text anywhere?

We thank the reviewer for pointing out this oversight, we have now added the figure reference in the revised manuscript. 

I have made a few more notes in the attached pdf and have highlighted some obvious grammar errors.

 We have now addressed these in the revised version where they occur.

---

## [Decision Letter · Decision Letter 1]

21 Jan 2020

PONE-D-19-26887R1

The fast and the frugal: Divergent locomotory strategies drive limb lengthening in theropod dinosaurs

PLOS ONE

Dear Dr. Dececchi,

Thank you for submitting your manuscript to PLOS ONE. After careful consideration, we feel that it has merit but does not fully meet PLOS ONE’s publication criteria as it currently stands. Therefore, we invite you to submit a revised version of the manuscript that addresses the points raised during the review process.

Both reviewers believe that you have made significant improvements to the first draft of the manuscript and have suggested minor revisions. However, they both have suggested you need to spend time to provide some confidence intervals to your estimates, with reviewer #1 detailing a good method for doing so. Please address this, and the discussion with regards to running. Do not rush this bit to get it done by the default revision due date, as I appreciate providing the confidence intervals for all taxa may take a bit of time.

We would appreciate receiving your revised manuscript by Mar 06 2020 11:59PM. To enhance the reproducibility of your results, we recommend that if applicable you deposit your laboratory protocols in protocols.io, where a protocol can be assigned its own identifier (DOI) such that it can be cited independently in the future. For instructions see: http://journals.plos.org/plosone/s/submission-guidelines#loc-laboratory-protocols

We look forward to receiving your revised manuscript.

Kind regards,

Andrew Cuff

Academic Editor

PLOS ONE

Reviewers' comments:

Reviewer's Responses to Questions

**Comments to the Author**

1. If the authors have adequately addressed your comments raised in a previous round of review and you feel that this manuscript is now acceptable for publication, you may indicate that here to bypass the “Comments to the Author” section, enter your conflict of interest statement in the “Confidential to Editor” section, and submit your "Accept" recommendation.

Reviewer #1: (No Response)

Reviewer #2: (No Response)

2. Is the manuscript technically sound, and do the data support the conclusions?

Reviewer #1: Partly

Reviewer #2: Yes

3. Has the statistical analysis been performed appropriately and rigorously? 

Reviewer #1: No

Reviewer #2: Yes

4. Have the authors made all data underlying the findings in their manuscript fully available?

Reviewer #1: Yes

Reviewer #2: Yes

5. Is the manuscript presented in an intelligible fashion and written in standard English?

Reviewer #1: Yes

Reviewer #2: Yes

6. Review Comments to the Author

Reviewer #1: Thank you to the authors for providing detailed and considered responses to our initial concerns. Through their inclusion of a new section on the potential confounding effects of body posture, I do believe the manuscript has been considerably improved. Likewise, the inclusion of a clear hypothesis has improved the framing of the study. Following up from Reviewer 2's previous suggestion, I do think two further additions are necessary in order for the research to be publishable:

1. Include confidence/prediction intervals at every step of the analysis. Although the authors are using volumetric mass estimates, many of those are published with an 'upper' and 'lower' bound solutions. Likewise, for any predictive model derived from empirical data (maximums speed equations of Alexandr or Ruiz and Torres, BMR estimates, CoT estimates from Pontzer), the specific model used should be clearly stated in text (including coefficients, r2 value etc) and 95% prediction intervals calculated. These can be easily calculated when the authors of the model have included their raw data (as is the case with Pontzer 2007, Grady et al etc.). Preferably, the authors should include this uncertainty as they progress through their analysis. This can be achieved through defining the 95% prediction interval and then running a random number generator within those bounds, say 1000 times. Progress those values on and repeat at the next stage of analysis. Ultimately, each species should then be represented by a distribution of values for predicted speed, CoT etc. This is a powerful approach, as it would then allow the authors to statistically test whether one species is different from another.

2. Following on from this, I would like to see the authors run their analysis on a few modern taxa to check the protocol produces sensible values. There are several volumetric models of extant animals available in the literature. Falkingham 2011 has a photogrammetry model of an elephant in the supplementary data, for example. I think the authors could have far more confidence in their results if they could demonstrate that they are capable of producing ballpark reasonable values for modern taxa for which we have the in vivo data.

Reviewer #2: Considering the revision and the authors' response, the authors do not seem to have addressed my largest two concerns.

The first is about the nature of the values calculated from the regressions. Consider, for example, the data from Table 2, where the authors argue for a clear difference between large bodied tyrannosauroids and other similarly sized theropods. Sinraptor and Giganotosaurus appear close to Tyrannosaur values, while Tarbosaurus is close to the Tyrannosaur values. Yes, the point estimates produce a pattern where Tyrannosaurs are generally below similarly-sized theropods but without an idea of the variance/error involved in estimation, it is unclear how strongly to interpret these results. I continue to believe that presenting point estimates for fossil taxa without any discussion or attempt to quantify variance, error, or confidence about these estimates is not nearly as helpful as if these were at included or at least considered. For instance, many of the large theropods exceed the largest taxon sampled in Pontzer's data (Obviously, since elephant is the largest available taxon with measured cost of transport data). Extrapolating beyond the sample set used to generate the regression is an issue that should be discussed in the paper. I think at least some discussion of these sorts of issues, even if the authors do not quantify them, is appropriate.

The second is that in a paper dealing with theropods and using the word "running" the authors should make explicit what they mean by the term. This is not a kinematics or energetics paper, but it is very easy for other researchers and the public to misinterpret any results that deal with running speed without things being made clear.

7. PLOS authors have the option to publish the peer review history of their article (what does this mean?). If published, this will include your full peer review and any attached files.

Reviewer #1: No

Reviewer #2: No

---

## [Author Response · Author response to Decision Letter 1]

10 Mar 2020

Review Comments to the Author

Reviewer #1: Thank you to the authors for providing detailed and considered responses to our initial concerns. Through their inclusion of a new section on the potential confounding effects of body posture, I do believe the manuscript has been considerably improved. Likewise, the inclusion of a clear hypothesis has improved the framing of the study. Following up from Reviewer 2's previous suggestion, I do think two further additions are necessary in order for the research to be publishable:

1. Include confidence/prediction intervals at every step of the analysis. Although the authors are using volumetric mass estimates, many of those are published with an 'upper' and 'lower' bound solutions. Likewise, for any predictive model derived from empirical data (maximums speed equations of Alexandr or Ruiz and Torres, BMR estimates, CoT estimates from Pontzer), the specific model used should be clearly stated in text (including coefficients, r2 value etc) and 95% prediction intervals calculated. These can be easily calculated when the authors of the model have included their raw data (as is the case with Pontzer 2007, Grady et al etc.). Preferably, the authors should include this uncertainty as they progress through their analysis. This can be achieved through defining the 95% prediction interval and then running a random number generator within those bounds, say 1000 times. Progress those values on and repeat at the next stage of analysis. Ultimately, each species should then be represented by a distribution of values for predicted speed, CoT etc. This is a powerful approach, as it would then allow the authors to statistically test whether one species is different from another.

We agree with this suggestion, and in trying to accommodate it we have provided confidence intervals based on running speed estimates derived from Hirt et al. (2017) (based on 1.96x the standard error values provided in Supplemental Table 4) as well as cost of transport based on Pontzer 2007 which provides the equation used in Pontzer et al. 2009 (which does not provide data to determine error bars). It should be noted that neither Alexander 1976 nor Ruiz 2018 (this was an error on our part, for while Ruiz and Torices 2013 does talk about running and does correct the error in Alexander 1976 Ruiz 2018 directly address the application of this error to a theropod trackway and thus is more applicable for use on line 143. This has been corrected in the text) include confidence intervals nor standard error values. Furthermore papers that use these speed estimates such as Thulborn 1982, Mazetta and Blanco 2001, McCrea et al. 2014, Smith et al. 2016 (amongst others) calculate them without confidence intervals. Thus we estimated speeds from those equations based both on the data presented and in the way that is acceptable and commonly used in the field. 

As for calculations from Froude numbers we do not include confidence intervals on those as the Froude number is a dimensionless parameter not a derived coefficient from a regression analysis and thus does not permit upper and lower bound estimates. For mass values, amongst the studies using volumetric reconstruction to estimate dinosaur size used in our analysis only those from Bates et al. (2009) include upper and lower bound estimates, the rest all include only a single value for calculated mass. Given that this represents only 2 data points in our analysis we do not believe that including them provides much added clarity or changes our results in any meaningful way. We have modified Figure 5 (include here as well) to document upper and lower bounds for the Hirt et al. 2017 speed values. We also calculated upper and lower bounds for CoT for the taxa included in Table 1and included these values as Sup. Table 6. 

Besides the inability to gain the data to perform the analysis this reviewer request, another challenge is that we do not believe that their suggested approach is a) informative and b) impactful on our results. While it may appear to enhance the reliability of the data (though as we pointed out it is not possible to do given the sources that we used) its true value is limited. Biologically speaking those “randomly” selected points are not actually equal probabilistically. For example, without any prior knowledge of suspected phylogenetic signal in differences in metabolic costs (which for the tyrannosaurs and the more basal theropods used in this part of the analysis we have no evidence nor suggestion that there would be any) assuming that it is equally likely that one clade occupies the upper bounds while the other the lower is not justified. In order to show that our data suggests that there are genuinely marked CoT differences, but not speed differences we have included comparisons of the Upper and lower bounds estimates for similar sized taxa between these different groups included here and as Supporting Figures 1 and 2 in the manuscript. What is of note is that the for similar sized taxa the difference in speed values are minute, but CoT are much larger, often by close to an order of magnitude. For example the difference between top speed ( all data based on mass reconstructions from Snively et al. 2019, using the Hirt et al. 2017 speed limitation upper bounds) between Giganotosaurus (MUCPv-CH-1) and Tyrannosaur rex (either CM 9380 or AMNH 5027) is 0.02m/s or about 0.29% of top speed values (~8.9m/s) while the CoT differences are 2.4 and 3.1% higher for the basal theropods. We see similar values in smaller exemplars such as Sinraptor vs Gorgosaurs (both individuals at ~2400kg) where a speed difference of 0.05 m/s (0.5% of top speed) is met with a 8.3% difference in CoT. This pattern persists if one looks at the lower bounds or mean values across comparisons. We believe by including this data we can clearly show a distinct pattern of markedly higher savings in CoT compared to minor differences in top speed values across the comparisons. Given that CoT is also a cumulative value which effects the energy budget every step that animal takes during its lifetime, this is likely a much more impactful value for selection.

2. Following on from this, I would like to see the authors run their analysis on a few modern taxa to check the protocol produces sensible values. There are several volumetric models of extant animals available in the literature. Falkingham 2011 has a photogrammetry model of an elephant in the supplementary data, for example. I think the authors could have far more confidence in their results if they could demonstrate that they are capable of producing ballpark reasonable values for modern taxa for which we have the in vivo data.

We are slightly puzzled by this request as both the volumetric estimates we present as well as all speed and CoT equations used were derived from studies and protocols tested first on extant animals. Thus the reason we can have some confidence in these protocols is because they have been derived from living creatures. What we can say is that perhaps that our Froude estimated speed potential for the largest taxa (mass >6000kg) may be too high as the fastest a living elephant has reliably been documented to run is Fr=3.4 (Hutchinson et al. 2003). This may be a moot point as using Hirt et al.’s upper bounds speed values we find that these correspond to Froude values at around 2.5 for these extremely large theropods. This is similar too (but higher than) values seen in modern African elephants on level tracks (Hutchinson et al. 2003), conditions that are unlikely to occur in the environments these animals existed in. Thus again we are confident that these values to likely represent upper estimates of speed for these creatures. 

Reviewer #2: Considering the revision and the authors' response, the authors do not seem to have addressed my largest two concerns.

The first is about the nature of the values calculated from the regressions. Consider, for example, the data from Table 2, where the authors argue for a clear difference between large bodied tyrannosauroids and other similarly sized theropods. Sinraptor and Giganotosaurus appear close to Tyrannosaur values, while Tarbosaurus is close to the Tyrannosaur values. Yes, the point estimates produce a pattern where Tyrannosaurs are generally below similarly-sized theropods but without an idea of the variance/error involved in estimation, it is unclear how strongly to interpret these results. I continue to believe that presenting point estimates for fossil taxa without any discussion or attempt to quantify variance, error, or confidence about these estimates is not nearly as helpful as if these were at included or at least considered. For instance, many of the large theropods exceed the largest taxon sampled in Pontzer's data (Obviously, since elephant is the largest available taxon with measured cost of transport data). Extrapolating beyond the sample set used to generate the regression is an issue that should be discussed in the paper. I think at least some discussion of these sorts of issues, even if the authors do not quantify them, is appropriate.

We addressed this issue above with the additions of new supporting tables and figures documenting more thoroughly the tyrannosaur values difference from other theropods of similar size. While we could include a small note about how the largest of our specimens extend beyond the modern taxa used to derived CoT equations unless there is reason to believe that one clade had a fundamentally different relationship between CoT and size than other how this would change our values. Even if our estimates for the largest theropods are off by 20% or more they should be off for all specimens by similar amounts, therefore not biasing our results one way or another and giving us the same overall conclusion (long legs are energetic savers in larger taxa) even if the caloric values we estimated are inaccurate. 

The second is that in a paper dealing with theropods and using the word "running" the authors should make explicit what they mean by the term. This is not a kinematics or energetics paper, but it is very easy for other researchers and the public to misinterpret any results that deal with running speed without things being made clear.

The authors are confused as to why this is an issue for the reviewer, and what implications it has for this publications. Nevertheless we have included the following section in the text to make it clear that we are using “running ability” to mean more than overcoming the walk to run transition barrier. 

“In extant vertebrates the walk to run transitions occurs at a Froude number > 1 [23], and a similar value is expected to hold for non-avian theropods with even the largest being suspected to achieve this feat [10, 11]. Following these parameters we define running ability here as the ability to achieve speeds corresponding to Froude numbers significantly higher than 1, as opposed to running capacity which is the ability to generate values of 1. We hypothesize that allometry will have significant consequences for running ability and that the selective weighting of top speed versus reducing energetic expenditures will vary across Theropoda concordant with changes in body size. We also hypothesize that amongst the largest theropods, > 1000kg, running ability, as assessed by top speed potential, will not be a significant factor in the influencing the relative level of elongation of the distal hindlimb, including in tyrannosaurs. “

---

## [Decision Letter · Decision Letter 2]

15 Apr 2020

The fast and the frugal: Divergent locomotory strategies drive limb lengthening in theropod dinosaurs

PONE-D-19-26887R2

Dear Dr. Dececchi,

We are pleased to inform you that your manuscript has been judged scientifically suitable for publication and will be formally accepted for publication once it complies with all outstanding technical requirements.

With kind regards,

Andrew Cuff

Academic Editor

PLOS ONE

Additional Editor Comments (optional):

Reviewers' comments:

Reviewer's Responses to Questions

**Comments to the Author**

1. If the authors have adequately addressed your comments raised in a previous round of review and you feel that this manuscript is now acceptable for publication, you may indicate that here to bypass the “Comments to the Author” section, enter your conflict of interest statement in the “Confidential to Editor” section, and submit your "Accept" recommendation.

Reviewer #1: All comments have been addressed

Reviewer #2: All comments have been addressed

2. Is the manuscript technically sound, and do the data support the conclusions?

Reviewer #1: Yes

Reviewer #2: (No Response)

3. Has the statistical analysis been performed appropriately and rigorously? 

Reviewer #1: Yes

Reviewer #2: (No Response)

4. Have the authors made all data underlying the findings in their manuscript fully available?

Reviewer #1: Yes

Reviewer #2: (No Response)

5. Is the manuscript presented in an intelligible fashion and written in standard English?

Reviewer #1: Yes

Reviewer #2: (No Response)

6. Review Comments to the Author

Reviewer #1: Thank you for taking the time to consider my comments. I believe the manuscript has been improved in the process. My concern throughout has always been that it felt like the reviewers were attempting to prove an already favoured hypothesis, rather than trying to disprove a null hypothesis. Hence my preference for identifying and incorporating as many possible sources of error through the workflow, and the notion of taking an extant species and walking it through the entire protocol as if it were a fossil taxon. I don't agree with the authors that there is nothing to be learned from this approach. Whilst these equations have indeed been derived from extant datasets, it is still valuable to reapply them to a few modern taxa, just to understand how the errors and uncertainty are compounded at each step. A volumetric mass estimate applied to a elephant skeleton will NOT correctly predict the mass of that animal, there will be error. Likewise, the CoT and speed equations will NOT exactly predict those of the elephant. The errors will mount up.

Reviewer #2: (No Response)

7. PLOS authors have the option to publish the peer review history of their article (what does this mean?). If published, this will include your full peer review and any attached files.

Reviewer #1: No

Reviewer #2: No

---

## [Editor Report · Acceptance letter]

22 Apr 2020

PONE-D-19-26887R2 

The fast and the frugal: Divergent locomotory strategies drive limb lengthening in theropod dinosaurs 

Dear Dr. Dececchi:

I am pleased to inform you that your manuscript has been deemed suitable for publication in PLOS ONE. Congratulations! Your manuscript is now with our production department. 

With kind regards,

on behalf of

Dr. Andrew Cuff 

Academic Editor

PLOS ONE